# MoTVLA: A Vision-Language-Action Model with Unified Fast-Slow Reasoning

## Abstract

Integrating visual-language instructions into visuomotor policies is gaining momentum in robot learning for enhancing open-world generalization. Despite promising advances, existing approaches face two challenges: limited language steerability when no generated reasoning is used as a condition, or significant inference latency when reasoning is incorporated. In this work, we introduce MoTVLA, a mixture-of-transformers (MoT)–based vision–language–action (VLA) model that integrates fast–slow unified reasoning with behavior policy learning. MoTVLA preserves the general intelligence of pre-trained VLMs (serving as the generalist) for tasks such as perception, scene understanding, and semantic planning, while incorporating a domain expert, a second transformer that shares knowledge with the pretrained VLM, to generate fast domain-specific reasoning (e.g., robot motion decomposition), thereby improving policy execution efficiency. By conditioning the action expert on decomposed motion instructions, MoTVLA can learn diverse behaviors and substantially improve language steerability. Extensive evaluations across natural language processing benchmarks, robotic simulation environments, and real-world experiments confirm the superiority of MoTVLA in both language reasoning and manipulation task performance. We refer to Project Page for the demonstration videos and corresponding descriptions.

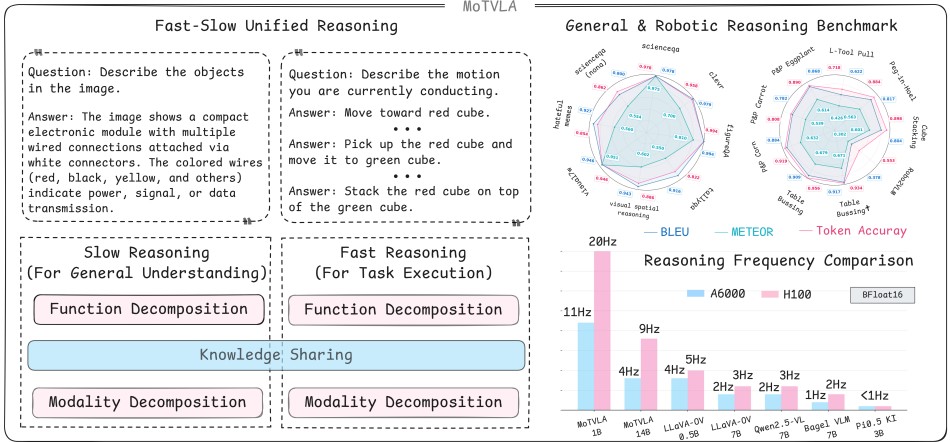

Figure 1: **Unified fast–slow reasoning in MoTVLA and its reasoning performance.** MoTVLA unifies fast–slow reasoning through a Mixture-of-Transformers architecture, in which the input modalities are first decomposed, followed by knowledge sharing, and finally decomposed again from a functional perspective. As a result, MoTVLA not only learns both general and domain-specific knowledge effectively, but also achieves superior reasoning efficiency.

# 1 Introduction

Vision-Language-Action (VLA) models, as natural extensions of Vision-Language Models (VLMs), have recently attracted growing interest in robot learning by generating action trajectories through

next-token prediction (Kim et al., 2025; Zawalski et al., 2025). While promising progress has been made in tasks such as manipulation (Zhao et al., 2025; Zitkovich et al., 2023) and mobile navigation (Cheng et al., 2025), this paradigm faces inherent limitations. Compared to natural language processing (NLP), robotics datasets are much smaller, and fine-tuning large VLAs on such limited data often degrades the general intelligence acquired during pre-training, reducing adaptability and generalization. Moreover, representing continuous actions with discretized tokens compromises precision and robustness (Driess et al., 2025).

To address these issues, diffusion policies (DPs) have emerged as a compelling alternative, better suited for modeling continuous action spaces. Leveraging their multimodal nature (Ho et al., 2020; Song et al., 2021), visuomotor DPs capture diverse behaviors via iterative noise–denoise processes (Chi et al., 2024). One paradigm integrates VLMs with DPs by conditioning the policies on separately encoded textual and visual features, enabling multitasking through multimodal inputs (Liu et al., 2025), though this design suffers from limited language steerability since reasoning is not explicitly generated and conditioned (Barreiros et al., 2025a). Another paradigm employs autoregressive VLMs to generate reasoning in the language domain, with DPs subsequently conditioned on this reasoning for action generation (Intelligence et al., 2025; Driess et al., 2025). While this approach improves behavioral generalization, inference remains constrained by the latency of next-token prediction, which hinders time-critical applications.

To bridge the aforementioned gap, we propose a mixture-of-transformers (MoT)–based (Liang et al.) vision-language-action (VLA) model, termed MoTVLA (*pronounced "MotiLA"*), which integrates fast–slow unified reasoning with policy learning through diffusion policies (DPs). Unlike existing approaches, MoTVLA unifies fast and slow reasoning within a single architecture by decomposing the modalities at the input and the functionalities at the output, while maintaining a shared global knowledge base in between (Fig. 1). The slow reasoning process follows the standard next-token prediction paradigm of the VLM (generalist), whereas the fast reasoning process is realized via token-wise prediction in a secondary transformer (domain expert) that shares global self-attention with the generalist. For illustration, we refer to these as the first and second transformers; however, in MoTVLA they are integrated into one architecture through a shared global attention mechanism. This design enables MoTVLA to retain the general intelligence of pre-trained VLMs for tasks such as perception, scene understanding, and semantic planning, while efficiently acquiring domain-specific reasoning, such as robot motion decomposition, at a faster pace. Finally, the action expert, implemented as a diffusion transformer (DiT), is conditioned on the reasoning signals together with visual and physical states to generate language-steered action trajectories, thereby closing the loop from high-level reasoning to low-level control.

We conduct comprehensive evaluations of MoTVLA across natural language processing (NLP) benchmarks, robotic simulation environments (ManiSkill) (Gu et al., 2023), and real-world experiments. These validations cover a wide spectrum of tasks, ranging from visual reasoning on image-based vision–question–answering (VQA) for text and mathematics to robotic manipulation tasks such as stacking, tool usage, insertion, pick-and-place, and table bussing. The results consistently confirm the superiority of MoTVLA over state-of-the-art (SOTA) baselines in both reasoning and manipulation tasks. Finally, the ablation study confirms the significance of the proposed architecture for both language reasoning and policy learning.

We summarize our main contributions as follows: (*i*) We unify fast and slow reasoning within a single model based on the MoT architecture, enabling the preservation of general intelligence while efficiently learning domain-specific knowledge that benefits from it; (*ii*) We condition policy learning on decomposed motion reasoning, thereby facilitating faster task execution while maintaining interpretability of policy behaviors within a language context. (*iii*) MoTVLA achieves superior performance in inference latency, language reasoning, and manipulation tasks, providing a novel insight of integrating reasoning into downstream behavior policy.

**Outline.** After a review of related work on VLA and DP in Section 2, we detail the MoTVLA model and training recipe in Section 3. We present experimental results in Section 4 and conclude in Section 5. Additional details, including inference latency comparisons, pseudo-code of the inference pipeline, dataset descriptions, and further qualitative results, are provided in the Appendix A and Supplementary Material.

## 2 RELATED WORK

**Vision-Language-Action Models.** Vision-language-action (VLA) models, as an important branch of robotic foundation models, have emerged as one of the most prominent approaches for learning multitask policies in robot learning. Owing to their billion-scale parameterization, VLAs are capable of accommodating large-scale robotic datasets (Khazatsky et al.; O'Neill et al., 2024; Walke et al., 2023; Ji et al., 2025; Chen et al., 2025a; Li et al., 2025b). The underlying rationale of this paradigm is to learn behavior policies through next-token prediction, analogous to language modeling, thereby transferring general intelligence into domain-specific knowledge for robotic tasks. Representative examples include RT-2 (Zitkovich et al., 2023) and OpenVLA (Kim et al., 2025), which pioneered the learning of visuomotor control policies by leveraging vision-language models (VLMs) and modeling continuous actions through discretized action tokens. Recognizing that purely end-to-end training can impair reasoning capabilities, subsequent work such as ECoT (Zawalski et al., 2025), Gemini Robotics (Team et al., 2025), and CoT-VLA (Zhao et al., 2025) proposed to jointly learn textual and visual reasoning alongside visuomotor policy learning. Despite their demonstrated success across diverse robotic tasks, VLAs face challenges that hinder their practical applications. In particular, control accuracy is often compromised by the information loss incurred when continuous actions are represented with discrete tokens (Driess et al., 2025).

**Diffusion Policy.** DPs (Chi et al., 2024; Xue et al., 2025) have been widely adopted for robot policy learning by leveraging the strong generative capabilities of diffusion models (Song et al., 2021; Ho et al., 2020) in visual generation. The central idea of DP is to model the multimodality of robot behaviors through the noise–denoise process, which is naturally suited for continuous action spaces. Recently, an emerging research direction has focused on advancing diffusion-based VLAs (Liu et al., 2025; Wen et al., 2025; Intelligence et al., 2025; Driess et al., 2025; Deng et al., 2025b; Chen et al., 2025b; Bjorck et al., 2025; Barreiros et al., 2025b) by integrating VLMs with DP, thereby combining the strengths of both paradigms. For example, RDT-1B (Liu et al., 2025) tokenizes textual and visual inputs, encodes them, and conditions the action diffusion process on this information, resulting in a multitask DP. However, as highlighted by LBM (Barreiros et al., 2025b), such lightweight integration faces limitations in language steerability because it only encodes input information, which inherently lacks reasoning content. In contrast, the $\pi 0.5$ family (Intelligence et al., 2025; Driess et al., 2025) first generates textual reasoning based on input images and prompts, and then conditions the DP on this reasoning through flow matching, thereby enabling instruction-following action policies. While these approaches achieve impressive generalization in real-world settings, their reliance on next-token prediction for reasoning introduces significant inference latency, which in turn limits task execution efficiency.

## 3 THE MOTVLA MODEL AND TRAINING RECIPE

### 3.1 MODEL ARCHITECTURE

The overall architecture of MoTVLA is illustrated in Fig. 2. MoTVLA adopts a Mixture-of-Transformers (MoT) design and consists of three key components: a generalist, a domain expert, and an action expert. The generalist is dedicated to visual–textual multimodal understanding, the domain expert focuses on fast reasoning for robotic tasks, and the action expert is responsible for multitask policy learning.

**Input Space Design.** The input modalities of MoTVLA consist of three domains: (1) language, which provides either general or domain-specific prompts, (2) RGB images, and (3) a set of learnable queries conditioned for fast reasoning generation. To process these inputs, MoTVLA employs a text tokenizer and a visual encoder that jointly support both fast and slow reasoning, along with learnable embeddings specifically designed for fast reasoning. Following BAGEL (Deng et al., 2025a), we adopt a Vision Transformer (ViT) as the visual encoder, initialized from SigLIP2-so400m/14 (Tschannen et al., 2025) with a fixed input resolution of 384. For the text tokenizer, we directly use the one from the pre-trained Qwen2.5 LLM (Hui et al., 2024).

**Reasoning Backbone Design: Decomposition-Composition-Decomposition.** To realize fast–slow unified reasoning, we follow the MoT design principle (Liang et al.). Specifically, we adopt Qwen2.5 LLM 7B (Hui et al., 2024) as the generalist backbone and mirror the same architecture for the

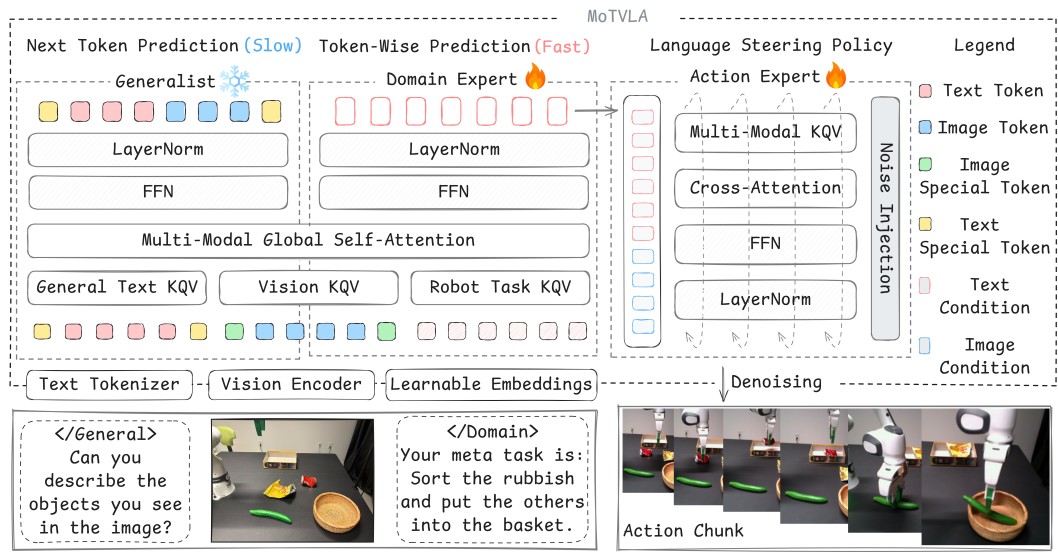

Figure 2: **The universal framework of MoTVLA.** MoTVLA adopts a Mixture-of-Transformers architecture comprising a generalist, a domain expert, and an action expert. Its reasoning backbone follows a decomposition–composition–decomposition pipeline: multimodal inputs are first processed independently, then integrated through a unified global self-attention mechanism, and finally decoupled at the output to perform slow and fast reasoning via the generalist and domain expert, respectively. The fast reasoning module decomposes robotic motions, and the resulting motion signals, together with visual and physical states, condition the action expert. This design ensures that the learned policy aligns with motion instructions and enhances language steerability, even under ambiguous prompts.

domain expert, which includes RMSNorm (Zhang & Sennrich, 2019), RoPE (Su et al., 2024) for positional embedding, and an additional QK-Norm module. Task information is decomposed into the three modalities introduced above and tokenized separately to produce multimodal tokens and their corresponding QKVs. These QKVs are then aggregated into a unified set for joint global attention, with modality-specific masks regulating interaction and conditioning, formulated as:

$$
\begin{aligned}
\text{GA}(x, \{\theta_{\text{LA}}^m\}) &= \left(\text{softmax}(\frac{QK^T}{\sqrt{d_k}})V\right)W_{QKV}^{m_i} \\
\text{att} &= \text{GA}(x, \{\theta_{\text{LA}}^m\}_{m \in \{\text{text, image, queries}\}}) \\
h &= x + \text{LayerNorm}_{\text{LA}}^{m_i}(\text{att}_i) \, ,
\end{aligned}
\tag{1}
$$

where GA and LA denote global and local attention, $x = x_1, x_2, ..., x_n$ is the input token sequence, $m$ represents the modality, $\theta$ the trainable parameters, and $W^{m_i}$ the modality-specific projection weights. In this formulation, QKVs of later modalities are allowed to attend to those of earlier ones. Within each modality, we apply two types of local attention for text and maintain bidirectional attention for vision. The global attention is then decomposed by modality indices, enabling distinct functions such as general and domain-specific reasoning. Following this decomposition–composition–decomposition paradigm, MoTVLA preserves the general intelligence inherited from pre-training by decoupling the associated parameters, while also facilitating domain-specific reasoning through effective knowledge sharing from the generalist to the domain expert. The importance of this design is validated by the ablation study in Section 4.3. Notably, the current design requires the generalist and domain expert to share the same model size, resulting in the MoTVLA reasoning backbone containing twice the parameters of the generalist alone. All primary training and evaluation in this work are conducted with MoTVLA-14B, while the 1B variant is used only to illustrate potential inference speed gains when scaling down. Limitations are discussed in Section 4.3, and inference frequency comparisons are provided in Appendix A.1.

**Reasoning Output Design.** The reasoning output of MoTVLA is unified in the textual space but decoupled into two functionalities: slow and fast reasoning. Slow reasoning follows the standard next-

token prediction paradigm with causal attention, leveraging the strengths of autoregressive LLMs but incurring high latency. Owing to large-scale pre-training on internet-scale datasets, the generalist demonstrates strong generalization across tasks such as perception, scene understanding, and semantic planning. In contrast, fast reasoning adopts a token-wise prediction paradigm with bidirectional attention, enabling substantially faster text generation. This design allows MoTVLA to pass hidden state inferences from the domain expert directly to the action expert without multiple forward passes. The key insight is that hidden states from a single forward process in token-wise prediction already encode the information required for reasoning generation, whereas those from next-token prediction reflect only the input information. Although token-wise prediction inevitably sacrifices some reasoning accuracy, it is sufficient for producing simple outputs such as decomposed manipulation motions in this work.

**Action Expert Design.** In our setting, to better accommodate the varying token lengths of hidden states and their associated masks, we adopt the Diffusion Transformer (DiT) as the action expert of MoTVLA, instead of a U-Net. Policy learning is performed within the framework of action diffusion (Chi et al., 2024), which captures the multimodal nature of robot behaviors. The state space of the action expert consists of four components: (1) visual observations $I_{t-H_I:t}$ with time horizon $H_I$, (2) semantic conditioning signals $h_{\ell_{DE}}$ generated by the domain expert, (3) the robot configuration $q_{t-H_I:t}$ (e.g., joint angles and gripper status), and (4) noisy action trajectories $A_{t:t+H_A}$, where $H_A$ denotes the action horizon. The diffusion policy learned by the action expert can be formulated as:

$$\pi_{\theta_{AE}}(A_{t:t+H_A}, h_{\ell_{DE}}|I_{t-H_I:t}, \ell, q_{t-H_I:t}) = \pi_{\theta_{AE}}(A_{t:t+H_A}|\mathcal{I}_{t-H_I:t}, h_{\ell_{DE}}, q_{t-H_I:t})\pi_{\theta_{RE}}(h_{\ell_{DE}}|I_t, \ell), \quad (2)$$

where $h_{\ell_{DE}}$ represents the hidden states of fast reasoning, $\ell$ denotes the input prompts, and $H_\mathcal{I}$ indicates the time horizon of visual observations. The parameters $\theta_{AE}$, $\theta_{DE}$, and $\theta_{RE}$ correspond to the trainable weights of the action expert, domain expert, and the reasoning backbone (comprising both the generalist and domain expert), respectively.

The full pseudo-code of the forward inference pipeline is presented in Appendix A.2.

## 3.2 TRAINING RECIPE

Following the abovementioned architecture, MoTVLA has three individual parts, the generalist, domain expert, and action expert, to train with. To leverage strong power of the general intelligence, in this work we adopt the pre-trained VLM from Bagel (Deng et al., 2025a), which achieves SOTA performance on visual understanding benchmarks, as the initialization of the generalist. Therefore, we dedicate our effort to train the rest of two stages in this work.

**Domain Expert Supervised Fine-Tuning.** In the domain expert SFT stage, we construct a high-quality manipulation motion VQA dataset by combining both simulated data and real-world demonstrations from human operators. For data construction, we adopt a question template of the form "Your meta task is: ..." and fill the subsequent part with a specific task description, e.g., "Sort the rubbish into the box and move the other into the basket." For the corresponding answer, we employ the generalist to generate decomposed motions in four steps, which are then used as the training labels for the action expert. To further improve generalization capability, we jointly train the action expert with two additional open-source datasets: LLaVA-OV (Li et al., 2024), which contributes to language generalization, and Robo2VLM (Chen et al., 2025a), which enhances robotic reasoning. We manually filter and curate long-answer samples from LLaVA-OV to ensure their suitability for token-wise prediction learning, while converting the selection-style annotations in Robo2VLM into reasoning-based labels. In total, our action expert reasoning dataset consists of 1.27M QA pairs, including 154K samples from simulation and 125K from real-world demonstrations collected in-house, 678K from Robo2VLM, and 318K from LLaVA-OV. Further details are provided in Appendix A.3. The learning objective is to minimize the negative log-likelihood of target tokens:

$$\mathcal{L}(\theta_{DE}) = \mathbb{E}_{(x,y)\sim D}[-\log p_\theta(y_{1:n}|x_{1:n})] \quad (3)$$

where $D$ denotes dataset and $y$ represents sequence of target reasoning tokens in this stage.

**Action Expert Diffusion Policy.** We collect 300 demonstrations in the ManiSkill (Gu et al., 2023) simulator for each of the three short-horizon tasks (Cube Stacking, Peg-in-Hole, and L-Tool Pull), with additional distractors introduced into the scenes. For the long-horizon tasks, we collect 100 demonstrations for each (Tool Pull & Place, Table Bussing, and Table Bussing Reverse). In the real

Table 1: Fast reasoning evaluation on both robotics and LLaVA-OV VQA tasks. ($*$ refers to revision, $\dagger$ denotes the same task evaluated under an alternative instruction prompt.)

| Task | Dataset | Metrics | | | |
|---|---|---|---|---|---|
| | | BLEU (0-1)↑ | CIDEr (0-10) ↑ | METEOR (0-1)↑ | Token Accuracy (%)↑ |
| Robotics | Cube Stacking | 0.8041 | 8.0574 | 0.6005 | 89.82 |
| | Peg-in-Hole | 0.8174 | 7.6805 | 0.5633 | 88.41 |
| | L-tool Pull | 0.6221 | 6.6255 | 0.4263 | 71.76 |
| | Pick & Place: Eggplant | 0.8680 | 8.3945 | 0.6136 | 88.99 |
| | Pick & Place: Carrot | 0.7816 | 7.1222 | 0.5395 | 80.83 |
| | Pick & Place: Corn | 0.8836 | 8.7838 | 0.6320 | 91.86 |
| | Table Bussing | 0.9086 | 8.9802 | 0.6794 | 95.59 |
| | Table Bussing† | 0.9168 | 9.0454 | 0.6712 | 93.41 |
| | Robo2VLM* | 0.3782 | 2.3601 | 0.3279 | 57.05 |
| LLaVA-OV VQA | FigureQA (Kahou et al., 2017) | 0.9940 | 2.4850 | 0.8105 | 99.40 |
| | CLEVR (Johnson et al., 2017) | 0.9790 | 2.3950 | 0.7004 | 95.80 |
| | ScienceQA (Saikh et al., 2022) | 0.9780 | 2.4450 | 0.9749 | 97.80 |
| | ScienceQA(nona context) (Li et al., 2024) | 0.8004 | 2.4282 | 0.5544 | 86.16 |
| | Hateful-memes (Kiela et al., 2020) | 0.9270 | 2.1350 | 0.5603 | 85.40 |
| | Visual-7W (Zhu et al., 2016) | 0.9459 | 2.3647 | 0.9510 | 94.59 |
| | Visual Spatial Reasoning (Liu et al., 2023) | 0.9430 | 2.2150 | 0.6022 | 88.60 |
| | TallyQA (Acharya et al., 2019) | 0.9160 | 2.0850 | 0.5498 | 83.20 |

world, using the SpaceMouse, we collect 50 demonstrations for pick-and-place with various vegetables and 200 demonstrations for table bussing. Notably, this dataset is substantially smaller than that used for training the domain expert, as publicly available datasets with decomposed motion annotations are hardly available, e.g. $\pi 0.5$ (Intelligence et al., 2025), and the workload of collecting and annotating such data in-house is prohibitively large. Conditioning on the visual observations $I_{t-H_I:t}$ over $H_I = 5$ observation horizons, robot configuration $q_{t-H_I:t}$, and decomposed motion reasoning signal $h_{\ell_{DE}}$, we implement a visuomotor policy using a conditional denoising diffusion model (Chi et al., 2024). During testing, the policy denoises Gaussian noise into action trajectories $A^0_{t:t+H_A}$ consisting of $H_A$ steps starting at time $t$. Specifically, beginning from Gaussian noise $A^K_{t:t+H_A}$, the denoising network $\epsilon_\theta$ iteratively refines the actions over $K$ denoising steps until producing the noise-free action $A^0_{t:t+H_A}$, computed as:

$$A^{k-1}_{t:t+H_A} = \bar{\alpha}_k \left( A^k_{t:t+H_A} - \epsilon_\theta \left( A^k_{t:t+H_A}, k, I_{t-H_I:t}, q_{t-H_I:t}, h_{\ell_{DE}} \right) \right) + \bar{\beta}_k \, \epsilon^k, \tag{4}$$

where $\epsilon^k \sim \mathcal{N}(0, \mathbf{I})$ and $\bar{\alpha}_k, \bar{\beta}_k$ are noise scheduler coefficients.

To train the denoising network $\epsilon_\theta$, we corrupt the ground-truth action $A^0_{t:t+Ha}$ with noise $\epsilon^k$ at the $k$-th step, and optimize the network to predict the injected noise (Chi et al., 2024), expressed as:

$$\mathcal{L}(\theta_{AE}) = \text{MSE} \left( \epsilon^k, \, \epsilon_\theta \left( \bar{\alpha}_k A^0_{t:t+H_A} + \bar{\beta}_k \epsilon^k, \, k, \, I_{t-H_I:t}, q_{t-H_I:t}, h_{\ell_{DE}} \right) \right). \tag{5}$$

## 4 EXPERIMENTS

MoTVLA is a multitasking model capable of performing general reasoning, multimodal robot-specific reasoning, and action planning for manipulation tasks. We conduct extensive evaluations in both simulation and real-world experiments, covering semantic reasoning as well as embodied action execution. It is important to note that for semantic reasoning we validate only the performance of fast reasoning, since the VLM component of standard reasoning is initialized from a pre-trained model, as detailed in the original work Deng et al. (2025a).

### 4.1 METRICS AND BASELINES.

In this work, we employ standard NLP metrics, including BLEU (Papineni et al., 2002), METEOR (Banerjee & Lavie, 2005), CIDEr (Vedantam et al., 2015), and token accuracy, to evaluate reasoning performance. For manipulation tasks, we report the average success rate using random seeds that were not observed during data collection or training.

We compare MoTVLA against several well-known and SOTA baselines, including transformer-based DP (Chi et al., 2024), GR-MG (Li et al., 2025a), $\pi 0$ (Black et al., 2024), and the recently released $\pi 0.5$ with knowledge insulation (Driess et al., 2025). For fair comparison and to unify the baselines' output space to our hardware setting, we fine-tuned them with our dataset.

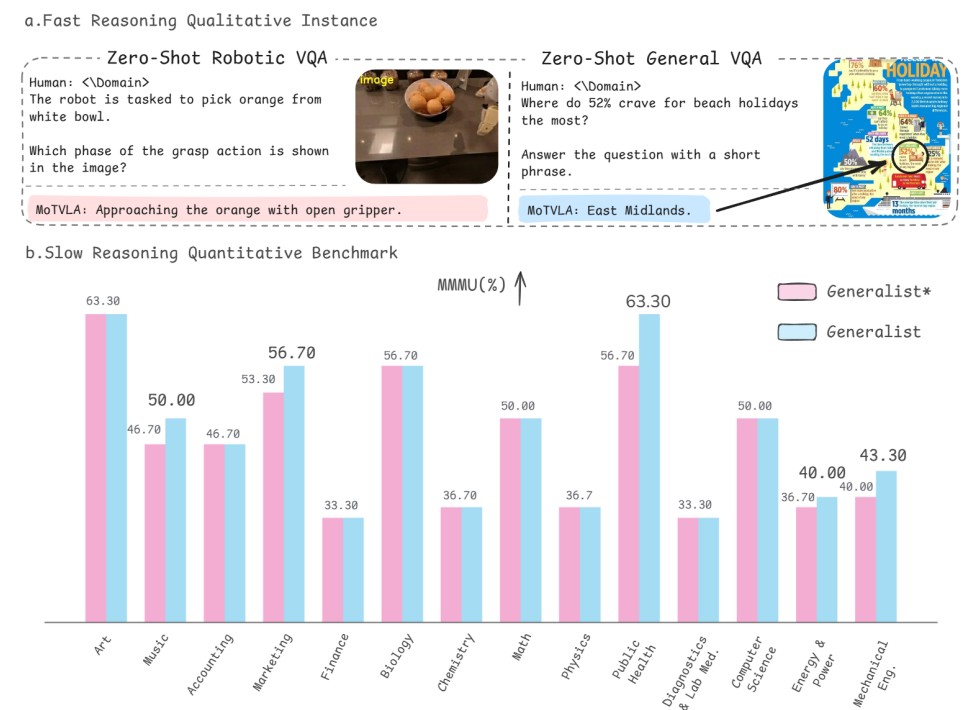

Figure 3: **Fast reasoning instances and slow reasoning benchmark.** ∗ indicates that the generalist is fine-tuned on the data originally intended for training the domain expert.

## 4.2 REASONING TASKS

The performance of fast reasoning is particularly critical in our work, since its outputs serve as conditioning signals for the action expert. A completely hallucinated reasoning would misguide the diffusion policy and lead to undesirable behaviors. As discussed in Section 3.2, we incorporate additional VQA datasets, namely LLaVA-OV (Li et al., 2024) and Robo2VLM (Chen et al., 2025a), in addition to our own robotic motion reasoning dataset, to enhance generalization capability. Distinct from existing VLA approaches, we not only qualitatively assess the quality of fast reasoning through VQA, but also quantitatively evaluate its performance on both robotic and general datasets.

**Analysis.** The overall statistical results of fast reasoning are summarized in Table 1. The average BLEU score and token accuracy across robotics and LLaVA-OV VQA tasks highlight its superior precision in capturing both domain expertise and common knowledge. Meanwhile, the CIDEr and METEOR scores on robotics tasks exhibit strong alignment with human judgment in terms of expression diversity and semantic similarity, indicating that the motion decomposition generated through fast reasoning has been effectively learned. The manually curated Robo2VLM reasoning dataset further underscores the difficulty of this evaluation, as we reformatted it from multiple-choice to reasoning-based VQA, encompassing diverse scenes, tasks, views, and reasoning domains (spatial, goal-oriented, and interaction-based). For LLaVA-OV VQA, although the CIDEr score is less prominent and below human-level quality, the relatively high METEOR score demonstrates sufficient generalization, justifying our motivation for joint training with both general and domain-specific datasets.

Qualitative evaluations on two randomly selected reasoning tasks (Fig.3a) further confirm strong zero-shot generalization. Surprisingly, MoTVLA is able to infer unseen objects and decompose motions never encountered during training, and can also extract key information from an information-dense poster. These results collectively demonstrate MoTVLA's strong domain knowledge and generalization capability to handle diverse scenarios. Detailed reasoning latency comparisons and more qualitative results are provided in Appendix A.1 and supplementary materials due to limited space.

Furthermore, we conduct an ablation study on slow reasoning for general knowledge evaluation by comparing training versus freezing the generalist, demonstrating the necessity of the MoT architec-

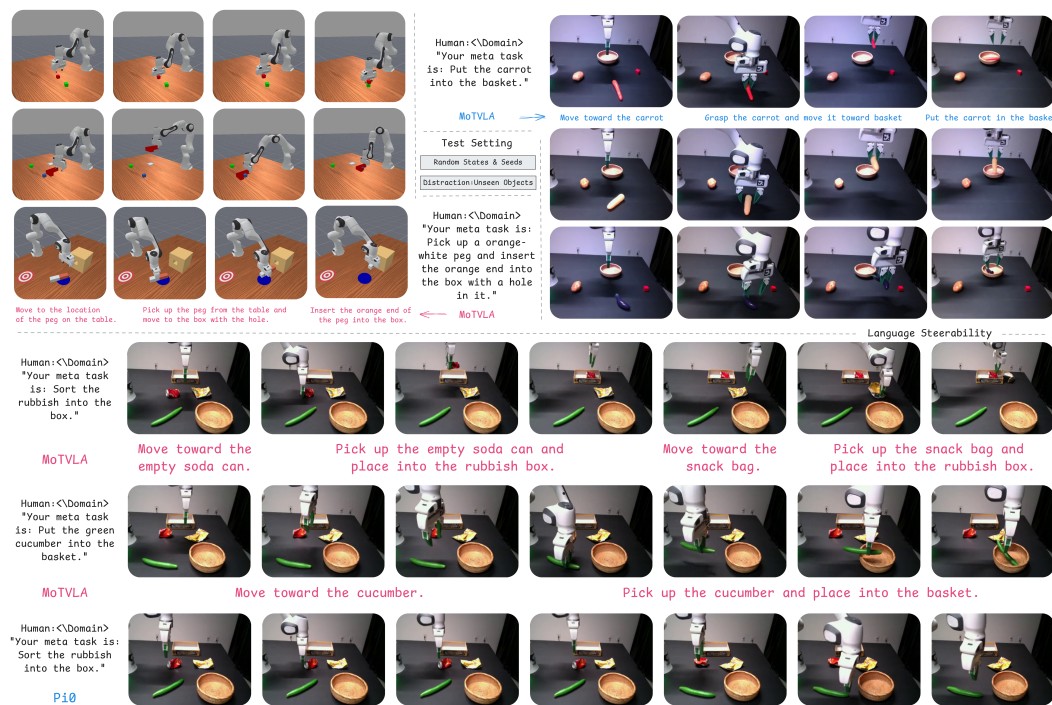

Figure 4: **Evaluation for manipulation tasks.** MoTVLA is rigorously evaluated in both simulation and real-world experiments. The testing suite encompasses diverse case types and variations, including ambiguous instruction prompts that require strong reasoning capability and language-steered behavioral policies.

ture in our approach. To this end, we fine-tuned the generalist using the data originally intended for training the domain expert and compared it with the unfine-tuned version. Since the generalist of MoTVLA is initialized from Bagel, we benchmark both versions on the Massive Multi-discipline Multimodal Understanding and Reasoning (MMMU) dataset (Yue et al., 2024), following the reasoning benchmark employed in the original Bagel paper. As shown in Fig. 3b, after fine-tuning, the performance of slow reasoning degrades across several subjects. For instance, while the tuned generalist maintains its performance in domains such as art, accounting, finance, and other science-related subjects, it exhibits knowledge forgetting in music, marketing, energy and power, and mechanical engineering. This degradation is even more pronounced in public health, where accuracy drops by 6.6% after fine-tuning. We also observe similar catastrophic forgetting in robotic VQA when providing a meta-task description followed by a general question, for example: "Your meta task is: Put the cucumber into the box. Before executing the task, tell me what is the color of the cucumber." The fine-tuned generalist fails to respond to the question about color and instead outputs the decomposed motion. These results highlight the superiority of MoTVLA, which preserves slow reasoning capability even after robotic VQA learning, confirming the necessity of the MoT architecture for maintaining fast–slow reasoning performance. Due to the space limitation, we refer the motion decomposition benchmark between fast reasoning of the MoTVLA and finetuned Bagel VLM 7B (generalist*) to the Appendix A.4.

Overall, the demonstrated superiority of fast reasoning performance establishes a solid foundation for MoTVLA to effectively learn robotic knowledge and corresponding behavior policies while maintaining its general intelligence. The policy learning aspect will be further analyzed, both qualitatively and quantitatively, in the following sections.

### 4.3 ROBOT MANIPULATION TASKS

We investigate whether MoTVLA can transform language-described motion decompositions into reliable manipulation across both simulation and real-world settings. Our objectives are threefold: (i)

Table 2: Performance comparison across different models for manipulation tasks.

| Short-horizon Tasks | MoTVLA | $\pi$0.5 KI | $\pi$0 | GR-MG | DP |
|---|---|---|---|---|---|
| Cube Stacking | **0.79** | 0.30 | 0.02 | 0.14 | 0.58 |
| Peg-in-Hole | **0.40** | 0.22 | 0.0 | 0.06 | 0.24 |
| L-Tool Pull | **0.62** | 0.36 | 0.28 | 0.10 | 0.48 |
| Pick & Place Eggplant | **1.0** | **1.0** | **1.0** | 0.50 | 0.25 |
| Pick & Place Corn | **1.0** | **1.0** | 0.75 | 0.0 | 0.50 |
| Pick & Place Carrot | **1.0** | **1.0** | **1.0** | 0.0 | 0.50 |
| Carrot w/ distractions | **1.0** | **1.0** | 0.75 | 0.0 | 0.75 |
| Eggplant w/ distractions | 0.75 | **1.0** | 0.75 | 0.0 | 0.50 |
| Corn w/ distractions | 0.75 | **1.0** | 0.50 | 0.0 | 0.50 |
| Mean ± Variance | **0.81** ± 0.04 | 0.76 ± 0.11 | 0.56 ± 0.11 | 0.09 ± 0.03 | 0.48 ± 0.02 |
| **Long-horizon Tasks** | | | | | |
| Tool Pull & Place | **0.72** | 0.10 | 0.08 | - | 0.50 |
| Table Bussing | **1.0** | 0.26 | 0.54 | - | 0.10 |
| Table Bussing Reverse | **0.98** | 0.70 | 0.50 | - | 0.60 |
| Mean ± Variance | **0.90** ± 0.02 | 0.35 ± 0.06 | 0.37 ± 0.06 | - | 0.40 ± 0.05 |
| **Action Training Recipe** | **Dataset and Scale** | | | | |
| fine-tuning | 1050 Collected Trajectories | | | | |
| Pre-training | None | 400h + much larger number data + OXE Intelligence et al. (2025) | $\simeq$ 10000h + OXE Black et al. (2024) | Ego4d (> 3500h) Li et al. (2025a) | None |

to assess whether motion-decomposed condition improves policy learning and robustness in contact-rich tasks, (ii) to examine whether decomposed motion snippets benefit language steerability of the policy behavior, and (iii) to thoroughly benchmark MoTVLA against four strong and SOTA baselines across an evaluation spectrum ranging from simulation to real-world experiments. To this end, we adopt multiple challenging short-horizon tasks with manually increased difficulty and long-horizon tasks with ambiguous instruction prompts that require strong internal reasoning, derived from the ManiSkill environment (Gu et al., 2023). In addition, two types of real-world experiments are conducted for training MoTVLA, including three single-stage manipulation tasks with clear instructions and a long-horizon multi-step task with ambiguous instructions.

During testing, these tasks are further diversified into additional cases by varying initial states, random seeds, and introducing unseen objects as distractions, as illustrated in Fig. 4. For example, MoTVLA must distinguish carrots, corn, and eggplants (targets) from potatos (distractions) in a zero-shot manner and guide the policy to complete the task accordingly. Full implementation details are provided in Appendix A.5.

**Analysis.** The quantitative comparison with baseline methods is reported in Table 2. As shown, MoTVLA consistently outperforms most baselines in both in-domain and zero-shot short-horizon tasks (with distractions), demonstrating strong robustness and generalization capability. Although the success rate on the challenging *Peg-in-Hole* task is relatively lower than that of other tasks due to its higher precision requirements and tight tolerances, MoTVLA still achieves the best performance among all methods. Furthermore, models with VLM backbones (MoTVLA, $\pi$0.5 KI, $\pi$0) consistently surpass those without pre-training, underscoring the importance of general intelligence in real-world robotic tasks. Last but not least, the dominant performance of MoTVLA on long-horizon tasks, which provide only ambiguous instructions and therefore require multiple steps of internal reasoning, further confirms the significance of motion decomposition and language steerability. Notably, while the $\pi$0.5 KI model performs significantly worse than MoTVLA on simulation tasks, it slightly surpasses our model in the real-world pick-and-place task. This result is reasonable, as $\pi$0.5 KI was fine-tuned from a model pre-trained on large-scale real-world robotic datasets, as indicated in Table 2. Interestingly, we observe that, unlike pick-and-place tasks, the $\pi$ series models struggle to complete tool-using tasks even after fine-tuning, regardless of whether they are short- or long-horizon. This phenomenon has also been reported in other studies (Qi et al., 2025).

The qualitative results presented in the lower part of Fig. 4 demonstrate the language steerability of MoTVLA compared with other baselines on the real-world table-bussing task, where only an ambiguous prompt, "Sort the rubbish into the box," was provided without specifying which objects constitute rubbish. As shown, MoTVLA successfully decomposes the task into several reasonable motions and guides the policy to complete the instruction, whereas $\pi_0$ fails in this task by treating all objects as rubbish and wandering back and forth among them. Moreover, we provide an alternative prompt in the same scene and observe that the policy behavior effectively aligns with the language

Table 3: **Ablation study of the architectural design.** P&P represents pick and place, TA denotes token accuracy, and SR refers to success rate.

| | | Bagel w/ DP | MoTVLA (scratch) | MoTVLA (ours) |
|---|---|---|---|---|
| Component | Generalist (Slow Reasoning) | Pre-trained | Scratch | Pre-trained |
| | Domain Expert (Fast Reasoning) | N/A | Fine-tuned | Fine-tuned |
| | Action Expert (Diffusion Policy) | Fine-tuned | Fine-tuned | Fine-tuned |
| Slow Reasoning | ScienceQA (TA) | 94.12 | 0.0 | 94.12 |
| | Motion Decomposition (TA) | 1.68 | 0.0 | 1.68 |
| Fast Reasoning | ScienceQA (TA) | N/A | 17.00 | 90.99 |
| | Motion Decomposition (TA) | N/A | 37.10 | 89.07 |
| Diffusion Policy | Tool Pull & Place (SR) | 0.0 | 0.62 | 0.72 |

instruction, thereby demonstrating the superiority of language steerability. Additional qualitative results on manipulation tasks are provided in supplementary materials.

**Ablation Study.** In this ablation study, we evaluate the impact of the pre-trained generalist and the necessity of the domain expert. We compare three variants: **Bagel (Deng et al., 2025a) w/ DP** who has only generalist and action expert pre-trained and fine-tuned respectively, enabling slow reasoning and diffusion policy but absent fast reasoning, **MoTVLA (scratch)** with randomly initialized the generalist and fine-tuned domain and action expert, thus able to perform both fast-slow reasoning and diffusion policy but lack general intelligence, and **MoTVLA (ours)** proposed by this work. The purpose of these baselines is twofold: (i) to investigate the significance of general intelligence by comparing our method with MoTVLA (scratch), and (ii) to illustrate the importance of the domain expert, which enables global attention, by comparing against Bagel w/ DP, where such global attention is absent. As shown in Table 3, MoTVLA (scratch) fails to learn across all tasks under both slow and fast reasoning, confirming that the general intelligence inherited from the pre-trained VLM is essential for effective multimodal reasoning. Meanwhile, Bagel w/ DP performs well on ScienceQA but achieves the lowest token accuracy in the slow reasoning of motion decomposition, indicating that the domain expert and its global attention mechanism are vital for aligning domain-specific knowledge and stabilizing motion generation. While it is possible to fine-tune the slow reasoning of Bagel on our motion decomposition dataset to improve accuracy, as discussed in Section 4.2, this approach would inevitably lead to catastrophic forgetting of general intelligence. Notably, MoTVLA significantly outperforms the other two baselines on long-horizon manipulation tasks, as the unstable and hallucinated motion decompositions produced by these baselines introduce substantial variance and hinder policy learning. Furthermore, the quantitative results of the ablation study on language steerability can be found in Appendix A.6.

**Limitations.** Despite its superior performance on most evaluation tasks, the full potential of MoTVLA has not yet been fully explored in this work. For example, we observe that inference speed can be significantly improved when both the generalist and domain expert are scaled down to 0.5B parameters. However, pre-training a 0.5B model to acquire general intelligence remains highly challenging, as it requires multiple stages of training with large-scale VQA datasets (Li et al., 2024; Hui et al., 2024). Moreover, the relatively limited amount of data available for training the action expert sometimes leads to strong reasoning ability but insufficient execution capability, resulting in failures on long-horizon tasks. This limitation could potentially be alleviated by scaling up the motion-annotated robotic data.

## 5 Conclusion

In this paper, we address the challenging gap between reasoning latency and language steerability by proposing MoTVLA, a MoT architecture–based robotic foundation model that unifies fast–slow reasoning and enhances the language steerability of behavior policies. MoTVLA acquires domain expertise and action policies through a two-stage curriculum learning scheme, while preserving the general intelligence inherited from a pretrained VLM even after the entire training process. Comprehensive benchmarking across language reasoning, simulation, and real-world experiments confirms the feasibility and superiority of the proposed approach. We believe that this novel integration of high-level reasoning and low-level control policies in VLA design paves the way for advancing robotic learning toward addressing open-ended language-instructed tasks and language-steered behaviors in large-scale open-world environments.

## 6 REPRODUCIBILITY STATEMENT

We are committed to ensuring reproducibility of our work. All code and scripts required to reproduce the results reported in this paper will be made publicly available upon acceptance of the paper.

## 7 THE USE OF LARGE LANGUAGE MODELS (LLMS).

The authors confirm that the entire content of this paper was conceived and written by themselves. Large language models were used only for language polishing.

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

# A    APPENDIX

## A.1    REASONING LATENCY.

Low reasoning latency is one of the advantage of the MoTVLA as it is able to perform fast reasoning with domain expert. We have thoroughly benchmarked the reasoning frequency of MoTVLA with two different sizes comparing with the $\pi0.5\_KI$, which has the similar motion reasoning capability but still within next token prediction paradigm, as well as several open-sourced VLMs. All models are evaluated with the same visual and textual inputs, with the maximum generation length restricted to 16 tokens and computations performed in bfloat16 precision. We measured the reasoning latency with 50 iterations for the same example to avoid the influence of cold start. As shown in Table 4, MoTVLA-14B significantly outperforms the 3B and 7B baselines on both H100 and A6000 GPUs. Moreover, when scaled down to 1B parameters, MoTVLA achieves a reasoning frequency nearly four times higher than that of LLaVA-OV-0.5B. It is important to note that MoTVLA-0.5B is used only to measure the inference latency of fast reasoning and has not yet been fully integrated into our framework, as pre-training a 0.5B generalist remains highly challenging and requires large-scale datasets. All quantitative and qualitative results reported in this paper are obtained with MoTVLA-14B.

Table 4: Comparison of reasoning latency.

| Model | Size | H100 | A6000 |
|---|---|---|---|
| $\pi0.5$ KI | 3B | < 1Hz | < 1Hz |
| Bagel-VLM | 7B | 2 Hz | 1 Hz |
| Qwen2.5-VL | 7B | 3 Hz | 2 Hz |
| LLaVA-OV | 7B | 3 Hz | 2 Hz |
| LLaVA-OV | 0.5B | 5 Hz | 4 Hz |
| MoTVLA | 14B | 9 Hz | 4 Hz |
| MoTVLA | 1B | 20 Hz | 11 Hz |

## A.2 INFERENCE PIPELINE

In this section, we describe the full inference pipeline of MoTVLA for both general reasoning and robotic manipulation tasks. MoTVLA adopts a fast–slow inference framework. At test time, the interaction proceeds in two modes: (i) the operator may engage in multi-turn dialogue with MoTVLA, posing questions or logical reasoning queries such as object descriptions or semantic planning for specific tasks, and (ii) the operator may instruct MoTVLA to execute manipulation tasks via task-specific prompts. This design not only ensures alignment between language and policy (e.g., when asked "What do you see in the image?" MoTVLA will answer the question without executing actions), but also enhances the interpretability of policy behaviors by generating intermediate motion decompositions during real-time inference. Figure 5 demonstrates a concrete instance of slow and fast reasoning during the inference time.

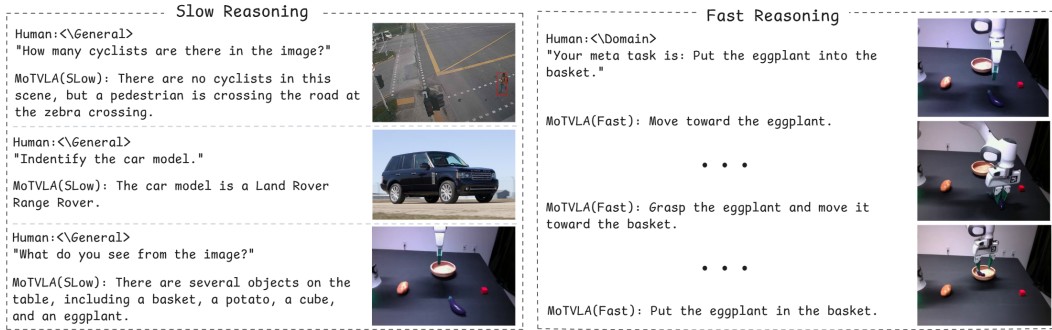

Figure 5: **A concrete example of fast and slow reasoning during inference.** When asked a general question, the generalist performs slow reasoning and provides a detailed verbal response. When a task-execution prompt is issued, the domain expert performs fast reasoning to generate decomposed intermediate motions for action execution.

During manipulation rollout, the system maintains sliding windows of the most recent $H_I$ images and robot states for context (line 10, line 11). At each step, the Domain Expert $DE$ performs fast, step-wise motion reasoning on the current image $I_t$ conditioned by the $task\_prompt$, producing the hidden state $h_{\ell_{DE}}$ (line 9). The Action Expert $AE$ then samples an action chunk $A^0_{H_A}$ conditioned on the image window $I_{t-H_I:t}$, state window $q_{t-H_I:t}$, and the hidden state $h_{\ell_{DE}}$ (line 12). Then action chunk $A^0_{H_A}$ is then sent to the ROBOTCONTROLLER for execution (line 13).

---

**Algorithm 1** MoTVLA Inference: Generalist $\rightarrow$ Domain Expert $\rightarrow$ Action Expert

---

**Input:** Models: $RE$ (Generalist/Reasoning backbone), $DE$ (Domain Expert), $AE$ (Action Expert)
**Input:** $I_0 \triangleright$ initial image, $\ell \triangleright$ user prompt, $H_I \triangleright$ obs-horizons, $H_A \triangleright$ action-horizons
**Output:** Executed closed-loop action sequence
 1: **procedure** MOTVLA_INFERENCE($I_0$, $\ell$)
 2:     IMGBUF $\leftarrow [\,]$
 3:     STATEBUF $\leftarrow [\,]$
 4:     $t \leftarrow 0$;
 5:     $general\_reasoning \leftarrow RE.$SLOWREASON($I_0$, $\ell$)             $\triangleright$ multi-turn dialogue
 6:     IMGBUF.PUSH($I_0$)
 7:     STATEBUF.PUSH(GETROBOTSTATE());
 8:     **while** $t <$ MAX_STEPS **do**
 9:         $h_{\ell_{\text{DE}}} \leftarrow DE.$FASTREASON($I_t$, $task\_prompt$)         $\triangleright$ fast, step-wise motion reasoning;
10:         $I_{t-H_I:t} \leftarrow$ IMGBUF.LAST($H_I$)
11:         $q_{t-H_I:t} \leftarrow$ STATEBUF.LAST($H_I$)
12:         $A^0_{t:t+H_A} \leftarrow AE.$SAMPLEDENOISE($I_{t-H_I:t}$, $q_{t-H_I:t}$, $h_t^{DE}$; horizon $= H_A$)
13:         ROBOTCONTROLLER.STEP($A^0_{t:t+H_A}$)
14:         $I_t \leftarrow$ GETCURRENTIMAGE()
15:         $q_t \leftarrow$ GETROBOTSTATE()                                 $\triangleright$ update observations
16:         IMGBUF.PUSH($I_t$)
17:         STATEBUF.PUSH($q_t$)                                          $\triangleright$ update buffer
18:         $t \leftarrow t + 1$
19:     **end while**
20:     STOPROBOT()
21: **end procedure**

---

## A.3 DATASET COMPOSITION.

Our SFT dataset for training the domain expert consists of 279.6K demonstrations collected in-house, 678K robotic data curated from Robo2VLM (Chen et al., 2025a), and 318K general knowledge samples open-sourced from the internet. The detailed composition of each dataset is summarized in Table 5.

Table 5: Composition of the dataset employed for supervised fine-tuning.

| Domain | Dataset | Size |
|---|---|---|
| Robotics | Cube Stacking | 32.2K |
| | Peg-in-Hole | 45.3K |
| | L-tool Pull | 76.9K |
| | Pick & Place: Eggplant | 11.0K |
| | Pick & Place: Carrot | 10.0K |
| | Pick & Place: Corn | 9.8K |
| | Table Bussing | 94.4K |
| | Robo2VLM* | 678.0K |
| General VQA | FigureQA | 100.0K |
| | CLEVR | 70.0K |
| | ScienceQA | 5.0K |
| | ScienceQA(nona context) | 19.2K |
| | Hateful-memes | 8.5K |
| | Visual-7W | 14.4K |
| | Visual Spatial Reasoning | 2.2K |
| | TallyQA | 98.7K |

## A.4 MOTION DECOMPOSITION BENCHMARK

In this section, we compare the reasoning performance of the domain expert in MoTVLA with that of the Bagel VLM 7B, which is fine-tuned on the same motion decomposition dataset as ours, as reported in Table 6.

Table 6: **Motion decomposition benchmark between domain expert of MoTVLA and finetuned Bagel VLM 7B.**

|  |  | BLEU | CIDEr | METEOR | Token Accuracy |
|---|---|---|---|---|---|
| Pick & Place: Eggplant | Finetuned Bagel VLM | 0.9741 | 8.9250 | 0.6385 | 89.40 |
|  | Domain Expert of MoTVLA | 0.8837 | 8.6153 | 0.6303 | 90.13 |
| Pick & Place: Carrot | Finetuned Bagel VLM | 0.8252 | 8.3441 | 0.5921 | 82.70 |
|  | Domain Expert of MoTVLA | 0.8435 | 7.9218 | 0.5835 | 88.47 |
| Pick & Place: Corn | Finetuned Bagel VLM | 0.8940 | 9.2152 | 0.6787 | 91.58 |
|  | Domain Expert of MoTVLA | 0.8081 | 7.8534 | 0.5785 | 82.53 |
| Table Bussing | Finetuned Bagel VLM | 0.9727 | 9.7389 | 0.7841 | 98.40 |
|  | Domain Expert of MoTVLA | 0.9018 | 8.6035 | 0.6678 | 92.47 |
| Table Bussing Reverse | Finetuned Bagel VLM | 0.9334 | 9.3886 | 0.7383 | 94.35 |
|  | Domain Expert of MoTVLA | 0.9086 | 8.9802 | 0.6794 | 95.59 |

From Table 6, it can be observed that the fine-tuned Bagel VLM 7B and the domain expert of MoTVLA each exhibit complementary strengths. However, we would like to mention that the performance of Bagel VLM is associated with the cost of suffering from catastrophic forgetting in terms of the general intelligence after the fine-tuning, as we discussed in Section 4.2.

A.5 TASKS AND MOTION ANNOTATION.

In this section, we elaborate on the demonstration tasks and corresponding motion annotation.

**Object relocation with stability (Cube Stacking).** The robot must pick up a red cube, stack it on a green cube, and release without the stack falling. *Motion decomposition:* (i) "Move toward the red cube." (ii) "Pick up the red cube." (iii) "Move it to green cube." (iv) "Stack the red cube on top of the green cube."

**Tight-tolerance insertion (Peg–in–Hole).** The robot must pick up an orange–white peg and insert the orange end into a box with a hole. To increase the difficulty, we manually augment the scene with distractions (a blue solid marker and a red dot marker). to the default setting. *Motion decomposition:* (i) "Move to the location of the peg on the table." (ii) "Pick up the peg from the table." (iii) "Move to the box with the hole." (iv) "Insert the orange end of the peg into the box."

**Tool-mediated manipulation beyond reach (L-tool Pull).** The robot must grasp an L-shaped tool and use it to pull a cube that lies beyond the arm's direct reach. To increase the difficulty, we manually augment the scene with distractions (a green sphere and a white socket). *Motion decomposition:* (i) "Move the robot's end effector to the position of the tool on the table." (ii) "Pick up the tool from the table." (iii) "Use the tool to pull the cube that is out of reach."

**Tool-mediated pull and place beyond reach (Tool Pull & Place).** The robot must grasp an L-shaped tool and use it to pull a cube located beyond the arm's direct reach. Afterward, it drops the tool to the side, picks up the cube, and places it at the designated destination. *Motion decomposition:* (i) "Move the robot's end effector to the position of the tool on the table." (ii) "Pick up the tool from the table." (iii) "Use the tool to pull the cube that is out of reach." (iv) "Move to the cube and pick it up." (v) "Place the cube into the bin."

**Table Bussing (Ambiguous Instruction).** The scene contains one *rubbish* item (an empty soda can), two *fruit* items (an apple and a banana), and two markers (ellipse). Since the instruction prompt does not explicitly specify which objects are rubbish, the model must interpret the instruction, ground the object categories, and generate a correct motion decomposition and execution order. *Ambiguous instruction:* "Place the garbage on the red ellipse and put all the other objects on the blue ellipse." *Motion decomposition:* (i) "Move to the can." (ii) "Pick up the can and place it on the red ellipse." (iii) "Move to the yellow banana." (iv) "Pick up the banana and place it on the blue ellipse." (v) "Move to the red apple." (vi) "Pick up the apple and place it on the blue ellipse."

**Table Bussing Reverse (Ambiguous Instruction).** The scene is the same as in the previous Table Bussing task; however, the execution order is reversed. *Ambiguous instruction:* "Place all the fruits on the blue ellipse and put the garbage on the red ellipse." *Motion decomposition:* (i) "Move to the red apple." (ii) "Pick up the apple and place it on the blue ellipse." (iii) "Move to the yellow banana." (iv) "Pick up the banana and place it on the blue ellipse." (v) "Move to the can. (vi) "Pick up the can and place it on the red ellipse."

**Real World Single-stage Pick-and-Place (Vegetables → Basket).** The robot must pick up a vegetable and place it into the basket. *Motion decomposition:* (i) "Move toward the {place_holder}." (ii) "Grasp the {place_holder} (iii) Move it toward the basket." (iv) "Put the {place_holder} in the basket." *Note:* {place_holder} corresponds to one of {corn, eggplant, carrot}; During the evaluation phase, we introduce three additional tasks containing distractions (a potato and a random cube) in the scene and assess the zero-shot performance.

**Real World Table Bussing (Ambiguous Instruction).** The scene contains two *rubbish* items (an empty soda can and a snack bag), one *other* item (a cucumber), a yellow box, and a basket. Since the instruction prompt does not explicitly specify which objects are rubbish, the model must interpret the instruction, ground the object categories, and generate a correct motion decomposition and execution order.

- *Ambiguous instruction:* "Sort the rubbish into the box." *Motion decomposition:* (i) "Move toward the empty soda can." (ii) "Pick up the empty soda can." (iii) "Place into the rubbish box." (iv) "Move toward the snack bag." (v) "Pick up the snack bag." (vi) "Place into the rubbish box."

```
Human:<\General>
"Is the peg inserted into the yellow
box? Provide the conclusion with a
simple reason.""

MoTVLA(SLow): No, the peg is not
inserted into the yellow box. The peg
is positioned outside the box,
resting on the surface of the table.
```

```
Human:<\Domain>
"Your meta task is: Pick up a orange-white peg and
insert the orange end into the box with a hole in
it."

MoTVLA(Fast): Move to the location of the peg on
the table.

            • • •

MoTVLA(Fast): Pick up the peg from the table.

            • • •

MoTVLA(Fast): Move to the box with the hole.

            • • •

MoTVLA(Fast): Insert the orange end of the peg into
the box.
```

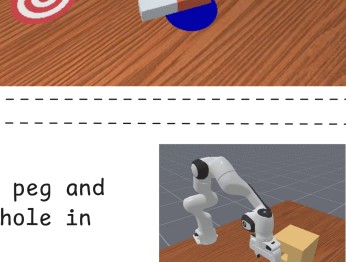

Figure 6: **A slow and fast reasoning annotation example of MoTVLA in the simulative environment.**

- *Clear instruction:* "Put the green cucumber into the basket." *Motion decomposition:* (i) "Move toward the cucumber." (ii) "Pick up the cucumber (iii) "Place it into the basket."

We collect 300 and 100 expert demonstrations for each short- and long-horizon task in simulation, respectively, 50 demonstrations for each real-world vegetable pick-and-place task (150 in total), and 100 demonstrations for each real-world table-bussing task (200 in total). Each demonstration is segmented into *motion decompositions* with aligned textual descriptions. We illustrate two concrete instances in Fig.6 and Fig.7. All training samples were collected using seeds ranging from 0 to the total number of trajectories (e.g., seeds 0–300 for the Peg-in-Hole task). Each seed triggers random-ization of the object positions on the table and initializes the Gaussian noise for the diffusion policy through the interface functions in our codebase. Furthermore, all baselines are trained on the iden-

Figure 7: **A slow and fast reasoning annotation example of MoTVLA in the real-world environment.**

tical datasets, with language input provided only to methods that initially support text conditioning. Notably, to ensure accurate and semantically consistent annotations while maintaining a relatively balanced number of samples for each motion type, we adopt the following motion merging strategy during the training. For pick and place tasks, we omit the "move toward" motion and merge "pick up" and "place into," since "place into" inherently contains the semantic meaning of "move toward."

Otherwise, "place into" would represent only a very short motion segment with an extremely small number of data samples. In contrast, for tasks involving more distinct terminating motions, such as "insert a peg into a hole" or "pull a cube with a tool," we retain the move toward" motion and merge it with pick up," as they are often performed simultaneously by the robotic arm, making it difficult to define a clear boundary between them, while keeping the final motion independent. This design helps mitigate potential overfitting to any dominant single motion pattern.

## A.6 LANGUAGE STEERABILITY OF MOTVLA.

To evaluate the language steerability of MoTVLA, we tested a checkpoint trained on scenes containing only a single object (either an apple or a banana). During inference, we placed both objects together and provided object-specific prompts to assess whether the model could correctly follow the instructions. For each test prompt, we conducted 20 trials using unseen seeds. The results are shown in Table 7.

Table 7: **Language steerability evaluation of MoTVLA.**

| Metric | Prompt: Put the red apple on the white ellipse. | Prompt: Put the yellow banana on the white ellipse. |
|---|---|---|
| Success Rate | 0.90 | 0.95 |

Although the visual zero-shot setup prevents MoTVLA from achieving perfect performance, the relatively high success rates demonstrate its strong language steerability, successfully picking and placing the instructed objects based on the given prompts. Additional qualitative results are provided on our website.

