# OpenReview forum: "MoTVLA: A Vision-Language-Action Model with Unified Fast-Slow Reasoning"
_ICLR.cc/2026/Conference — Submitted to ICLR 2026_

### Official Review · Reviewer_LEWC · 2025-10-22

**Soundness:** 3
**Presentation:** 4
**Contribution:** 2
**Rating:** 4
**Confidence:** 4

**Summary:**

The paper proposes a MoT based VLA model to address the challenging gap between reasoning lantency and language sterrability. It conducts VQA and Manipulation experiments to verify its model's capability.

**Strengths:**

The writing is clear, and the paper is exceptionally well organized. The figures are thoughtfully designed, with harmonious colors and a clean, balanced layout.

**Weaknesses:**

I doubt the necessity of introducing such large models (generalist and domain expert) merely to perform manipulation tasks like pick and place. As I understand it, the current work aims to achieve an integration of high-level reasoning and low-level control, but with the introduction of such a large number of parameters, I would expect to see tasks that cannot be accomplished by basic VLAs, in order to truly highlight the role of the generalist—rather than just demonstrating VQA-like capabilities, which are useless in actual manipulation. I believe the following types of tasks could better reflect the value of MoT based design, such as long-horizon tasks (where the “slow system” handles instruction decomposition and subtask transition) or zero-shot generalization to reflect the slow system reasoning ability.

**Questions:**

1. As I understand it, the generalist and domain expert modules are connected through a shared latent feature space, rather than by having the generalist’s output directly feed into the domain expert. Moreover, the generalist itself is not trained—it mainly serves to provide original pretrained vision-language model (VLM) features. Is my understanding correct?

2. In Table 1, there is no comparison with existing baselines, so I cannot assess how strong the VQA capability actually is.

3. In Table 3, I noticed that the domain expert runs at 9 Hz, while the generalist runs at only 2 Hz. My understanding is that the domain expert has roughly the same number of parameters as the generalist, plus an additional action expert module. Is the frequency difference due to the computation involving action chunks during inference?

---

> ### Author Response · Authors · 2025-11-22
> **Response to Reviewer LEWC (I)**
>
> Thanks for the reviewer’s appreciation regarding the clearness and presentation of our work! We have carefully gone through all the comments mentioned by the reviewer and we would like to provide a concrete point-by-point response in the following to address reviewer’s concern.
>
> ---
>
> ### **A1: I doubt the necessity of introducing such large models (generalist and domain expert) merely to perform manipulation tasks like pick and place. As I understand it, the current work aims to achieve an integration of high-level reasoning and low-level control, but with the introduction of such a large number of parameters, I would expect to see tasks that cannot be accomplished by basic VLAs, in order to truly highlight the role of the generalist—rather than just demonstrating VQA-like capabilities, which are useless in actual manipulation. I believe the following types of tasks could better reflect the value of MoT based design, such as long-horizon tasks (where the “slow system” handles instruction decomposition and subtask transition) or zero-shot generalization to reflect the slow system reasoning ability.**
>
> Thank you for the comments. We would like to address this concern with the following detailed clarifications and additional supporting experiments.
>
> First and foremost, we would like to clarify that the tasks conducted in this work are not limited to pick-and-place. The Peg-in-Hole and L-Tool Pull tasks shown in Table~2 are entirely different types of tasks, which explains the lower success rate compared to the others due to their higher level of difficulty. Nevertheless, we acknowledge that MoTVLA has been primarily verified on short-horizon tasks, and it would indeed be more convincing to evaluate its performance on long-horizon tasks with ambiguous prompts that require strong reasoning capabilities for motion decomposition.
>
> To address this, we have conducted three long-horizon tasks and compared the performance of MoTVLA against other baselines, as summarized below.
>
> **Note: To be continued in the next thread due to the characteristics limitations.**

---

> ### Author Response · Authors · 2025-11-22
> **Response to Reviewer LEWC (II)**
>
> ### **A1 (Continued)**
>
> To address this, we have conducted three long-horizon tasks and compared the performance of MoTVLA against other baselines, as summarized below.
>
> **Tool Pull and Place:**
>
>   1. **Prompts**: Your meta task is: Put the cube into the bin.
>   2. **Motion Decomposition:**
>      a) Move to the position of the L shaped tool on the table.
>      b) Pick up the L shaped tool from the table.
>      c) Use the L shaped tool to pull the cube.
>      d) Move to the cube and pick it up.
>      e) Place the cube into the bin.
>    3. **Description:**  In this task, the primary objective of MoTVLA is to use the tool on the table to pull a cube that is initially out of the reach of the manipulation arm and place it into the designated destination. The key challenge lies in the fact that, instead of providing a detailed instruction, only a simple prompt is given, i.e., “Put the cube into the bin,” which contains no information about the use of the tool. Consequently, the manipulation arm must infer and learn how to utilize the tool by following the motion decompositions. The task is considered successful if the cube is placed in the correct position within a fixed number of steps; otherwise, it is regarded as a failure.
>
>  **Table Bussing:**
>
>   1. **Prompts**: Your meta task is: Place the garbage on the red ellipse and put all the other objects on the blue ellipse. (Note: red/blue ellipse are the markers)
>   2. **Motion Decomposition:**
>     a)	Move to the can.
>     b)	Pick up the can and place it on the red ellipse.
>     c)	Move to the yellow banana.
>     d)	Pick up the banana and place it on the blue ellipse.
>     e)	Move to the red apple.
>      f) Pick up the apple and place it on the blue ellipse.
>    3. **Description:**  In this task, the primary objective of MoTVLA is to clear the objects on the table and classify them into their corresponding destinations. The key challenge lies in the fact that the user does not specify which objects are garbage and which are not. The robot must therefore infer this information autonomously and complete the task by following the decomposed motions rapidly inferred by MoTVLA. The task is considered successful if all objects are placed in their correct positions within a fixed number of steps and considered as a failure otherwise.
>
>  **Table Bussing Reverse:**
>
>   1. **Prompts**: Your meta task is: Place all the fruits on the blue ellipse and put the garbage on the red ellipse.
>   2. **Motion Decomposition:**
>     a)	Move to the red apple.
>     b)	Pick up the apple and place it on the blue ellipse.
>     c)	Move to the yellow banana.
>     d)	Pick up the banana and place it on the blue ellipse.
>     e) Move to the can.
>     f) Pick up the can and place it on the red ellipse.
>    3. **Description:**  This task is a variant of the Table Bussing task. The primary objective is to evaluate the language steerability of MoTVLA when provided with a reversed instruction prompt (from garbage  others to others  garbage), while the visual information remains largely similar. As before, the robot is not explicitly informed which objects are garbage and which are not; instead, MoTVLA must infer this information and complete the task autonomously. The task is determined successful when all the objects are placed to the correct position within fixed steps and failure otherwise.
>
> The following table presents the performance comparison of MoTVLA against the other three baselines on the aforementioned three tasks. All experiments are conducted under the same training settings and evaluated over 50 unseen seeds, with object positions and noise distributions of diffusion policy randomly initialized.
>
> | Model | Tool Pull & Place | Table Bussing| Table Bussing Reverse |
> |:------------|:-----------:|:------------:|:--------:|
> | DP | 0.50 | 0.10 | 0.60 |
> | $\pi$0 | 0.08| 0.54 | 0.50 |
> | $\pi$0.5 KI | 0.10 | 0.26 | 0.70 |
> | MoTVLA | **0.72** | **1.0** | **0.98** |
>
> It is evident that MoTVLA significantly outperforms the other baselines by a large margin across all three tasks. These results highlight the importance of motion decomposition (fast reasoning) in achieving language-steerable behaviors, particularly in long-horizon tasks with ambiguous instructions that demand strong reasoning capabilities. The corresponding demonstration video can be found in [Click Me! Tool Pull & Place](https://drive.google.com/file/d/1guNmHbDjpkuCnn4HJo2THY1soK46G9TN/view); &emsp; [Click Me! Table Bussing](https://drive.google.com/file/d/17vXHEUtTpo3aoK9P_k3rJ6upkHbRtvrM/view);  &emsp; [Click Me! Table Bussing Reverse](https://drive.google.com/file/d/1W0OVgwimJaNy6red0OEuypsHwCB8fJ2P/view).
>
> We will add quantitative results of long-horizon tasks as well as the corresponding analysis to the revised manuscript, and we hope this could well address your concerns.

---

> > ### Comment · Reviewer_LEWC · 2025-11-24
> >
> > Why do π₀.₅ KI and π₀ perform worse than DP in the toll place and pull tasks?
> > From my experience, π₀.₅ and π₀ typically outperform DP by a wide margin on these tasks, so this discrepancy requires a clear explanation. If this degradation indeed occurs, I would suspect that π₀.₅ and π₀ were not pretrained on sufficient simulator data, which could naturally lead to weaker performance.
> >
> > However, without a clear explaination, readers will doubt the accuracy of baseline implementation.

---

> > > ### Author Response · Authors · 2025-11-24
> > > **Re: Reviewer LEWC Re:Response to Reviewer LEWC (II)**
> > >
> > > Thank you to the reviewer for pointing out this important issue! **We agree that providing a clear explanation for the performance comparison would help readers better understand the reason behind the unexpected performance discrepancy.**
> > >
> > > As the reviewer correctly noted, we believe that one of the primary causes of the degraded performance in the Tool Pull & Place task stems from the **mismatch in dataset domains (the real-to-sim gap)**, since the $\pi$-series models are pre-trained solely on large-scale real-world datasets rather than simulated ones. This observation is consistent with the results in our manuscript (Table 2), where $\pi$0 and $\pi$0.5 KI also underperform across all three simulated tasks. We would like to highlight that our model and the DP baseline were not pre-trained on simulation data either. Since all models were trained under exactly the same settings, we believe that the comparison remains fair.
> > >
> > > Notably, **we fine-tuned only one model for each method to learn all three long-horizon tasks simultaneously**, following the official fine-tuning codebase provided by Physical Intelligence. As shown in the quantitative results, although the $\pi$-series models exhibited worse performance than DP on the Tool Pull & Place task, the same models outperformed DP on the other two tasks, achieving relatively high success rates. This consistency further confirms that our fine-tuning implementation for the $\pi$-series models is correct.
> > >
> > > An interesting phenomenon we observed is that the $\pi$-series models excel particularly in pick-and-place style tasks (e.g., Pick & Place Vegetables) but perform less reliably on manipulation tasks that involve more complex geometry or long-horizon reasoning, such as the Tool Pull & Place task. A similar phenomenon has been reported in other studies (e.g., Table I and Table III in [1]), suggesting that this issue is likely attributed to the pre-training recipe (datasets) or the model itself.
> > >
> > > Last but not least, to further enhance transparency and reproducibility, we are willing to open-source all of our fine-tuned checkpoints for $\pi$, $\pi$0.5 KI, DP, and MoTVLA on Hugging Face, if necessary.
> > >
> > > We thank again the reviewer's insighful comment and will incorporate the above analysis into the revised manuscript to improve reader understanding.
> > >
> > > #### **Reference**
> > > 1. Han Qi et al. “Compose by Focus: Scene Graph-based Atomic Skills” https://arxiv.org/pdf/2509.16053

---

> ### Author Response · Authors · 2025-11-22
> **Response to Reviewer LEWC (III)**
>
> ### **A2: As I understand it, the generalist and domain expert modules are connected through a shared latent feature space, rather than by having the generalist’s output directly feed into the domain expert. Moreover, the generalist itself is not trained—it mainly serves to provide original pretrained vision-language model (VLM) features. Is my understanding correct?**
>
> Yes, that is correct. MoTVLA leverages general intelligence by allowing the domain expert to attend to the generalist through connections across each attention layer, rather than treating it as an input. Regarding the generalist, we did not perform additional training, as the pre-trained model already achieves SOTA performance on multiple benchmarks such as MMMU and MathVista.
>
> ---
>
> ### **A3: In Table 1, there is no comparison with existing baselines, so I cannot assess how strong the VQA capability actually is.**
>
> Thank you to the reviewer for pointing out this concern. Indeed, we also attempted to identify fair baselines for motion decomposition in robotic reasoning; however, it is quite challenging to find an appropriate baseline that can generate sub-task motions while simultaneously producing actions conditioned on them, as our approach does. Notably, $\pi$0.5 KI is the most suitable baseline. Unfortunately, as stated on its official GitHub repository, the currently released version of $\pi$0.5 KI does not support the language head.
>
> While it would be possible to employ open-source VLMs for comparison, a fairer comparison would be to evaluate our generalist against these VLMs. To this end, we report a comparison between MoTVLA and two strong baselines, LLaVA-OV 7B and Qwen2.5-VL 7B, across multiple benchmarks, including MMBench [1], MMMU [2], MM-Vet [3], MathVista [4], and CODA-LM [5].
>
> | Model | MMBench | MMMU | MM-Vet | MathVista |CODA-LM|
> |:------------|:-----------:|:------------:|:-------:|:--------:|:--------:|
> | LLaVA-OV 7B | 80.8 | 48.8 | 57.5 | 63.2 | 1.3 |
> | QwenVL 2.5 7B | 83.5 | **58.6** | 67.1 | 68.2 | 5.1 |
> | MoTVLA VLM 7B | **85.0** | 55.3 | **67.2** | **73.1** | **6.4** |
>
> It is clear that the MoTVLA outperforms the other two well-known strong baselines on four tasks among five, confirming its SOTA performance of MoTVLA on slow reasoning.
>
> In addition, we have fine-tuned the Bagel VLM 7B on our robotic dataset and evaluated on the test set. The results are shown in table below.
>
> | Dataset | Model | BLEU | CIDEr | METEOR | Token Accuracy |
> |:------|:--------|:--------:|:--------:|:--------:|:--------:|
> | Pick & Place: Eggplant| Finetuned Bagel VLM 7B | 0.9741 | 8.9250 | 0.6385 | 89.40 |
> |  | MoTVLA Domain Expert 7B | 0.8837 | 8.6153 | 0.6303 | **90.13** |
> | Pick & Place: Carrot | Finetuned Bagel VLM 7B | 0.8252 | 8.3441 | 0.5921 | 82.70 |
> |  | MoTVLA Domain Expert 7B | 0.8435 | 7.9218 | 0.5835 | **88.47** |
> | Pick & Place: Corn | Finetuned Bagel VLM 7B | 0.8940 | 9.2152 | 0.6787 | **91.58** |
> |  | MoTVLA Domain Expert 7B | 0.8081 | 7.8534 | 0.5785 | 82.53 |
> | Table Bussing | Finetuned Bagel VLM 7B | 0.9727 | 9.7389 | 0.7841 | **98.40** |
> |  | MoTVLA Domain Expert 7B | 0.9018 | 8.6035 | 0.6678 | 92.47 |
> | Table Bussing Reverse | Finetuned Bagel VLM 7B | 0.9334 | 9.3886 | 0.7383 | 94.35 |
> |  | MoTVLA Domain Expert 7B | 0.9086 | 8.9802 | 0.6794 | **95.59** |
>
> From the table, it can be observed that the fine-tuned Bagel VLM 7B and the domain expert of MoTVLA each exhibit complementary strengths.
>
> **Note: To becontinued in the next thread due to characters limitation.**

---

> ### Author Response · Authors · 2025-11-22
> **Response to Reviewer LEWC (IV)**
>
> ### **A3 (Continued): In Table 1, there is no comparison with existing baselines, so I cannot assess how strong the VQA capability actually is.**
>
> However, we would like to mention that the performance of Bagel VLM is associated with the cost of suffering from catastrophic forgetting regarding the general intelligence after the fine-tuning.
>
> To verify it, we also compare the finetuned Bagel VLM 7B against with the generalist of our MoTVLA over the MMMU benchmark, illustrated in this [bar plot (Click Me!)](https://drive.google.com/file/d/1LFaGdrlfUuwdPu6N6cELKkzxYaqxf_hX/view). As shown in this figure, after fine-tuning, the performance of Bagel VLM 7B degrades across several subjects. For instance, while the tuned Bagel VLM 7B maintains its performance in domains such as art, accounting, finance, and other science-related subjects, it exhibits knowledge forgetting in music, marketing, energy and power, and mechanical engineering. This degradation is even more pronounced in public health, where accuracy drops by 6.6% after fine-tuning. We also observe similar catastrophic forgetting in robotic VQA when providing a meta-task description followed by a general question, for example: “Your meta task is: Put the cucumber into the box. Before executing the task, tell me what is the color of the cucumber.” The fine-tuned VLM fails to respond to the question about color and instead outputs the decomposed motion. These results highlight the superiority of MoTVLA, which preserves slow reasoning capability even after robotic VQA learning, confirming the necessity of the MoT architecture for maintaining fast–slow reasoning performance.
>
> We will include the above discussion in the revised manuscript and hope it adequately addresses the reviewer’s concern regarding the significance of preserving general intelligence.
>
> #### **Reference**
> 1. Liu, Yuan, et al. "Mmbench: Is your multi-modal model an all-around player?." European conference on computer vision. Cham: Springer Nature Switzerland, 2024.
> 2. Yue, Xiang, et al. "Mmmu: A massive multi-discipline multimodal understanding and reasoning benchmark for expert agi." Proceedings of the IEEE/CVF Conference on Computer Vision and Pattern Recognition. 2024.
> 3. Yu, Weihao, et al. "Mm-vet: Evaluating large multimodal models for integrated capabilities." arXiv preprint arXiv:2308.02490 (2023).
> 4. Lu, Pan, et al. "Mathvista: Evaluating mathematical reasoning of foundation models in visual contexts." arXiv preprint arXiv:2310.02255 (2023).
> 5. Li, Yanze, et al. "Automated evaluation of large vision-language models on self-driving corner cases." CoRR (2024).
>
>
> ---
>
> ### **A4: In Table 3, I noticed that the domain expert runs at 9 Hz, while the generalist runs at only 2 Hz. My understanding is that the domain expert has roughly the same number of parameters as the generalist, plus an additional action expert module. Is the frequency difference due to the computation involving action chunks during inference?**
>
> Thank you for your question. The frequency difference between the generalist and the domain expert arises from their distinct reasoning paradigms. The generalist employs next-token prediction, generating answers sequentially by predicting language tokens one at a time (needs multiple forward process), **whereas the domain expert adopts token-wise prediction, which decodes all tokens into a complete sentence in a single step.** Although token-wise prediction involves certain accuracy trade-offs compared to next-token prediction, we found it sufficient for motion decomposition tasks, where the responses are relatively simple and short. Consequently, the inference speed of the domain expert is considerably faster than that of the generalist, even though both possess the same number of parameters.
> Regarding the inference frequency of the action expert, for generating an action trunk consisting of 16 actions within 50 denoising steps, the rate is approximately 4 Hz on an NVIDIA A6000 GPU. we provide an inference latency comparison against CogACT [1], which employs the same type of action encoder as MoTVLA, and OpenVLA [2], a well-known closed-loop VLA.
>
> | GPU A6000 | MoTVLA | CogACT [1] | OpenVLA [2] |
> |:------------|:-----------:|:------------:|:--------:|
> | Inference Time (ms) | 252 | 181 |307 |
> |Denoising Steps | 50 | Unkown | N/A |
> | Number of generated actions | 16 | 16 | 1 |
>
> Although the comparison is not entirely fair due to differences in parameter sizes and denoising steps, we aim to provide a relevant intuition regarding the inference latency among VLAs. Thank you once again for raising this point and we hope this explanation clarifies your concerns.
>
> #### **Reference**
> 1. Li, Qixiu, et al. "Cogact: A foundational vision-language-action model for synergizing cognition and action in robotic manipulation." arXiv preprint arXiv:2411.19650 (2024).
> 2. Kim, Moo Jin, et al. "OpenVLA: An Open-Source Vision-Language-Action Model." Conference on Robot Learning. PMLR, 2025.

---

> ### Author Response · Authors · 2025-11-25
> **Update to Reviewer LEWC**
>
> We have uploaded the revised version of the PDF file, and we would be more than happy to hear any further feedback or engage in discussion with the reviewer to further enhance the quality of our work!

---

### Official Review · Reviewer_beNN · 2025-10-28

**Soundness:** 3
**Presentation:** 3
**Contribution:** 3
**Rating:** 6
**Confidence:** 3

**Summary:**

The paper proposes MoTVLA, a mixture-of-transformers vision–language–action model that unifies fast and slow reasoning to improve both language steerability and real-time performance in robot control. It retains a pretrained VLM “generalist” for slow, autoregressive reasoning (perception, semantic planning) and introduces a shared-knowledge “domain expert” that performs fast, token-wise motion decomposition to condition a diffusion-based action expert (DiT) for continuous control. Experiments across NLP/VQA benchmarks, ManiSkill simulations, and real robots show superior reasoning metrics and higher task success rates versus strong baselines (e.g., π0/π0.5 KI, DP), with markedly lower reasoning latency. Ablations confirm that both the pretrained generalist and the fast domain expert are critical for stable reasoning and policy success.

**Strengths:**

- Originality: The unified fast–slow reasoning architecture via a Mixture-of-Transformers is a creative synthesis that removes a key limitation in prior VLA systems—either poor steerability without explicit reasoning or high latency with autoregressive CoT—by sharing global attention between a pretrained generalist and a token-wise domain expert. The decomposition–composition–decomposition design and the use of fast motion decomposition to condition diffusion policies is a novel and pragmatic formulation.

- Quality: The paper presents a thorough empirical evaluation across complementary fronts (robotic simulation, real-world manipulation, and general VQA), with strong, consistent gains over strong baselines (π0/π0.5 KI, DP, GR-MG) and clear latency advantages.

- Clarity and Significance: The model components and inference pipeline are clearly described, supported by pseudo-code, figures, and metric choices that align with the intended behaviors.

**Weaknesses:**

- Switching between the slow and fast models is not yet autonomous. The generalist is used only for optional high-level reasoning or dialogue at the outset. During execution, it can be activated only by the operator.
- Ablations are conducted on a simple cube-stacking task, where explicit reasoning appears less necessary.

**Questions:**

- Does the model exhibit error-recovery behavior? If so, is additional annotation required?

---

> ### Author Response · Authors · 2025-11-22
> **Response to Reviewer beNN (I)**
>
> We greatly thank the reviewer for appreciating the novelty and experimental thoroughness of our work. We have thoroughly gone through the concerns raised by the reviewers and we would like to address them point by point with the detailed response together with the additional experiments.
>
> ---
>
> ### **A1: Switching between the slow and fast models is not yet autonomous. The generalist is used only for optional high-level reasoning or dialogue at the outset. During execution, it can be activated only by the operator.**
>
> Indeed, whether MoTVLA conducts dialogue or performs concrete tasks depends on the operator’s instruction. However, we would like to emphasize that this property is one of MoTVLA’s advantages rather than a limitation. Although MoTVLA is capable of activating both experts (the generalist and the domain expert) simultaneously, we intentionally design it to activate only one expert at a time according to the user’s instruction, as we believe that the foundation of language steerability lies in human-centric interaction. Consequently, MoTVLA can accurately respond to the operator’s intent, engaging in semantic dialogue when the user requests detailed answers to questions, or performing manipulation tasks when explicitly instructed to do so.
>
> This design is one of the key characteristics distinguishing MoTVLA from other SOTA VLAs, which often produce unnecessary outputs learned from training templates even when not prompted. A concrete example is when the user provides the prompt “Describe the objects you observe on the table”, a reasonable response would be to describe the relevant contents in textual form, rather than elaborating on sub-task planning and internal reasoning (ECoT [1]), or executing unintended actions ($\pi$0 [2]).
>
> #### **Reference**
> 1. Zawalski, Michał, et al. "Robotic Control via Embodied Chain-of-Thought Reasoning." Conference on Robot Learning. PMLR, 2025.
> 2. Black, Kevin, et al. "π0: A vision-language-action flow model for general robot control. CoRR, abs/2410.24164, 2024. doi: 10.48550." arXiv preprint ARXIV.2410.24164.
>
> ---
>
> ### **A2: Ablations are conducted on a simple cube-stacking task, where explicit reasoning appears less necessary.**
>
> This is a great point. It would indeed be more convincing to conduct ablation studies on long-horizon tasks with ambiguous instruction prompts, which would further highlight the necessity and significance of preserving the general intelligence of slow reasoning while learning domain knowledge through fast reasoning. We are currently performing ablation studies on one of the long-horizon tasks and will provide the results in a separate follow-up thread. Thank you for your patience!
>
> ---
>
> ### **A3: Does the model exhibit error-recovery behavior? If so, is additional annotation required?**
>
> Yes, it does! The error-recovery behavior naturally emerges from the generalization capability of our approach without requiring any additional annotations.
>
> More specifically, we observed that MoTVLA exhibits error-recovery behaviors in both simulation and real-world tasks, owing to the domain randomization technique used during training, where the initial noise distribution and object positions are varied. For example, MoTVLA repeatedly attempts to pick up objects (e.g., vegetables) during the pick-and-place task and retries pulling the cube if it fails to do so in the L-Tool Pull task. We have attached the corresponding demos [Click Me!:Gripper-Reclose](https://drive.google.com/file/d/1c7Tw991VhM4GYM6qYRALK1c5E4D7SR87/view), [Click Me!:Re-Pickup](https://drive.google.com/file/d/1Af_OSwxqtlggWzV9Bm03ZCcU_bxQhj5n/view), and [Click Me!:Re-pull](https://drive.google.com/file/d/12BobIp4vtuI-fkPMti9YHsknY1Qmsn4u/view).
>
> Please kindly check them out and we hope our response can well address your concerns!

---

> ### Author Response · Authors · 2025-11-25
> **Update to Reviewer beNN**
>
> We have uploaded the revised version of the PDF file, and we would be more than happy to hear any further feedback or engage in discussion with the reviewer to further enhance the quality of our work!

---

> ### Author Response · Authors · 2025-12-02
> **Response to Reviewer beNN Regarding A2**
>
> ### **A2: Ablations are conducted on a simple cube-stacking task, where explicit reasoning appears less necessary.**
>
> #### **Previous Response:**
>
> This is a great point. It would indeed be more convincing to conduct ablation studies on long-horizon tasks with ambiguous instruction prompts, which would further highlight the necessity and significance of preserving the general intelligence of slow reasoning while learning domain knowledge through fast reasoning. We are currently performing ablation studies on one of the long-horizon tasks and will provide the results in a separate follow-up thread. Thank you for your patience!
>
> #### **Update:**
> We appreciate the reviewer’s patience. We have obtained updated long-horizon task results for MoTVLA as well as the other baselines for the ablation study, and Table 3 in the manuscript has been revised accordingly.
>
> | Attribute | Attribute | Bagel w\ DP | MoTVLA (scratch) | MoTVLA (ours) |
> |:------|:--------|:--------:|:--------:|:--------:|
> | Component | Generalist (Slow Reasoning) | Pre-trained | Scratch | Pre-trained |
> |  | Domain Expert (Fast Reasoning) | N/A | Fine-tuned | Fine-tuned |
> |  | Action Expert (Diffusion Policy) | Fine-tuned | Fine-tuned | Fine-tuned |
> | Slow Reasoning| ScienceQA (TA) | 94.12 | 0.0 | 94.12 |
> |  | Motion Decomposition (TA) | 1.68 | 0.0 | 1.68 |
> | Fast Reasoning| ScienceQA (TA) | N/A | 17.00 | **90.99** |
> |  | Motion Decomposition (TA) | N/A | 37.10 | **89.07** |
> | Diffusion Policy | Tool Pull & Place (SR) | 0.0 | 62.0 | **72.0** |
>
> We have updated this table and corresponding analysis in the revised manuscript (Section 4.3 Ablation Study & Table 3) and we hope this could well address the reviewer's concern!

---

### Official Review · Reviewer_NduJ · 2025-10-30

**Soundness:** 2
**Presentation:** 2
**Contribution:** 2
**Rating:** 2
**Confidence:** 5

**Summary:**

MoTVLA proposes a Mixture-of-Transformers vision-language-action model that unifies general and robotic domain reasoning with behavior policy learning to balance language steerability and inference latency. The model is trained via SFT for the domain expert and diffusion policy learning for the action expert, and is evaluated on NLP benchmarks, ManiSkill simulations, and real-world manipulation.

**Strengths:**

S1. Using bidirectional token-wise reasoning can effectively accelerate the inference process.

S2. This paper conducts benchmarks on both general-domain and robotic-domain reasoning to evaluate their respective performances.

**Weaknesses:**

W1. **How are slow reasoning and fast reasoning defined?** I don’t find it convincing that general-domain reasoning is inherently slow while robotic-domain reasoning is fast. Moreover, reasoning in the robotic domain is more crucial for manipulation. Wouldn’t adopting token-wise prediction potentially lead to insufficient reasoning capability?

W2. **The paper lacks both motivation and ablation studies.** It does not verify why and how general and robotic-domain reasoning improve manipulation performance. Which aspect of manipulation is enhanced (e.g., fine-grained control, long-horizon planning, or generalization)? How can the authors justify that preserving science or other general reasoning abilities helps manipulation? This contradicts common understanding in robotics, where retaining such reasoning only increases parameter redundancy and inference latency for real-time control systems. Finally, during policy inference, the DiT head relies solely on fast reasoning features (plus visual observations, etc.) and does not use outputs from slow reasoning as conditions.

W3. **What is the real innovation of the MoT design compared with Bagel (Deng et al., 2025a)?** Is it merely distinguishing general-domain and robotic-domain datasets based on Bagel’s MoT structure and then adding a DiT policy head? Furthermore, the use of bidirectional token-wise reasoning is not motivated by characteristics of robotic tasks.

W4. As a robotic VLA paper, the model’s inference speed should be compared with more VLA baselines such as Fast [1], Pi_0, and CogACT [2]. Although these models do not explicitly generate language, the latent features provided by their LLMs also serve as high-level conditions for action generation.
[1] Fast: Efficient Action Tokenization for Vision-Language-Action Models
[2] CogACT: A Foundational Vision-Language-Action Model for Synergizing Cognition and Action in Robotic Manipulation

**Questions:**

Q. I think the authors need to clarify the innovation of the MoT design and the overall method design, as well as clearly explain the motivation of the paper, e.g., why and how general-domain reasoning can benefit robotic manipulation, and what exact aspects of manipulation it is expected to improve.

---

> ### Author Response · Authors · 2025-11-22
> **Response to Reviewer NduJ (I)**
>
> Thanks for the reviewers appreciation of superiority in terms of inference latency and thoroughness of evaluation over multiple benchmarks. We have thoroughly checked through all the comments and would like to respond to them point-by-point in the following.
>
> ---
> ### **A1: How are slow reasoning and fast reasoning defined? I don’t find it convincing that general-domain reasoning is inherently slow while robotic-domain reasoning is fast. Moreover, reasoning in the robotic domain is more crucial for manipulation. Wouldn’t adopting token-wise prediction potentially lead to insufficient reasoning capability?**
>
> Thanks for the reviewer’s questions. The distinction between slow and fast reasoning is defined by the prediction paradigm rather than by the general or robotic domain. As elaborated in **Section 3.1 Reasoning Output Design**, the generalist employs a next-token prediction paradigm, in which answers are generated sequentially based on the previous context. This approach requires multiple LLM forward passes and leads to incremental inference time as the answers become more detailed and lengthy. In contrast, the domain expert adopts a token-wise prediction paradigm, which can decode the entire answer within a single forward pass using the subsequent language head. Consequently, these distinct prediction paradigms result in substantially different inference latencies.
>
> We use the generalist to answer general questions and the domain expert to perform motion decomposition because the dialogue between the robot and the user needs to be detailed and comprehensive, requiring strong reasoning capability. In contrast, as discussed in **Section 3.1 Reasoning Output Design**, although token-wise prediction inevitably sacrifices some reasoning accuracy, it is sufficient for generating concise outputs such as decomposed manipulation motions in this work. Moreover, its low inference latency greatly benefits the efficiency of subsequent task execution. We hope this explanation well addresses the reviewer’s concerns.

---

> ### Author Response · Authors · 2025-11-22
> **Response to Reviewer NduJ (II)**
>
> ### **A2: The paper lacks both motivation and ablation studies. It does not verify why and how general and robotic-domain reasoning improve manipulation performance. Which aspect of manipulation is enhanced (e.g., fine-grained control, long-horizon planning, or generalization)? How can the authors justify that preserving science or other general reasoning abilities helps manipulation? This contradicts common understanding in robotics, where retaining such reasoning only increases parameter redundancy and inference latency for real-time control systems. Finally, during policy inference, the DiT head relies solely on fast reasoning features (plus visual observations, etc.) and does not use outputs from slow reasoning as conditions.**
>
> We would like to highlight that, as clearly stated in the Introduction section, our motivation is to preserve general intelligence while efficiently learning domain-specific knowledge that benefits from it. This design also aims to enhance task execution efficiency and improve the interpretability of concrete behaviors. For instance, imagine a humanoid robot operating in a household environment. General intelligence is essential when the human engages in dialogue with the robot without requesting any physical actions, whereas domain-specific intelligence becomes critical when the robot is required to perform a physical task assigned by the human, such as bringing a cup of coffee. Both modes are indispensable for developing a truly general-purpose robot.
>
> Below, we provide point-by-point responses to the reviewer’s comments.
>
>  **1. Which aspect of manipulation is enhanced?**
>
>   1. First and foremost, the scene understanding and reasoning capabilities of manipulation have been significantly enhanced. While most existing VLAs evaluate their performance only at the manipulation task level (primarily using success rate), we train and evaluate MoTVLA on both reasoning and manipulation tasks. As shown in Table 1 of our manuscript, the language metrics (BLEU, CIDEr, and METEOR) and absolute token accuracy demonstrate the superior reasoning capability of the domain expert, particularly in robotic VQA tasks.
> Furthermore, since MoTVLA preserves both general intelligence and domain-specific knowledge, it can accurately respond to the user’s intent. For example, when the user provides the prompt “Describe the objects you observe on the table,” ECoT[1] produces unnecessary outputs learned from the training labels, even when sub-task planning and internal reasoning are unrelated to the question. Similarly, Pi0[2] does not respond to the question at all but instead directly executes actions without being prompted. In contrast, MoTVLA can not only engage in dialogue with the user when a general question is asked but also execute task-specific actions when requested. This level of language steerability is essential, as VLAs should be human-centric and able to respond precisely to user instructions.
>
>   2. Secondly, the manipulation performance across diverse short-horizon tasks has been improved. Table2 in our manuscript illustrates the superiority of MoTVLA compared with four strong and SOTA baselines. We observe that both MoTVLA and the baseline VLAs perform relatively well on pick-and-place tasks, as these tasks are straightforward and require limited reasoning capability. However, for tasks that differ from pick-and-place and demand deeper internal reasoning, MoTVLA significantly outperforms the others.
> For instance, in the L-Tool Pull task, the VLA must reason that in order to pull the cube located out of the manipulator’s reach, it should pick up the tool by the appropriate handle so that the tool can extend to the cube’s position. A similar level of reasoning is required in another challenging task, Peg-in-Hole, where most baseline VLAs show relatively poor performance, while MoTVLA achieves the highest success rate thanks to its superior reasoning capability.
>
>   3. Additionally, we realize that it would be more convincing if MoTVLA is quantitatively evaluated over long-horizon tasks which require multi steps internal reasonings (motion decompositions), especially when the instruction prompt is ambiguous. To this end, we have conducted three long-horizon experiments as follows:
>
> **Note: To be continued in next thread due to characters limitation**
>
> #### &emsp; **Reference**
> 1. Zawalski, Michał, et al. "Robotic Control via Embodied Chain-of-Thought Reasoning." Conference on Robot Learning. PMLR, 2025.
> 2. Black, Kevin, et al. "π0: A vision-language-action flow model for general robot control. CoRR, abs/2410.24164, 2024. doi: 10.48550." arXiv preprint ARXIV.2410.24164.

---

> ### Author Response · Authors · 2025-11-22
> **Response to Reviewer NduJ (III)**
>
> ### **A2(Continued)**
>
> 3. (Continued) ... To this end, we have conducted three long-horizon experiments as follows:
>
> **Tool Pull and Place:**
>
>   1. **Prompts**: Your meta task is: Put the cube into the bin.
>   2. **Motion Decomposition:**
>      a) Move to the position of the L shaped tool on the table.
>      b) Pick up the L shaped tool from the table.
>      c) Use the L shaped tool to pull the cube.
>      d) Move to the cube and pick it up.
>      e) Place the cube into the bin.
>    3. **Description:**  In this task, the primary objective of MoTVLA is to use the tool on the table to pull a cube that is initially out of the reach of the manipulation arm and place it into the designated destination. The key challenge lies in the fact that, instead of providing a detailed instruction, only a simple prompt is given, i.e., “Put the cube into the bin,” which contains no information about the use of the tool. Consequently, the manipulation arm must infer and learn how to utilize the tool by following the motion decompositions. The task is considered successful if the cube is placed in the correct position within a fixed number of steps; otherwise, it is regarded as a failure.
>
>  **Table Bussing:**
>
>   1. **Prompts**: Your meta task is: Place the garbage on the red ellipse and put all the other objects on the blue ellipse. (Note: red/blue ellipse are the markers)
>   2. **Motion Decomposition:**
>     a)	Move to the can.
>     b)	Pick up the can and place it on the red ellipse.
>     c)	Move to the yellow banana.
>     d)	Pick up the banana and place it on the blue ellipse.
>     e)	Move to the red apple.
>      f) Pick up the apple and place it on the blue ellipse.
>    3. **Description:**  In this task, the primary objective of MoTVLA is to clear the objects on the table and classify them into their corresponding destinations. The key challenge lies in the fact that the user does not specify which objects are garbage and which are not. The robot must therefore infer this information autonomously and complete the task by following the decomposed motions rapidly inferred by MoTVLA. The task is considered successful if all objects are placed in their correct positions within a fixed number of steps and considered as a failure otherwise.
>
>  **Table Bussing Reverse:**
>
>   1. **Prompts**: Your meta task is: Place all the fruits on the blue ellipse and put the garbage on the red ellipse.
>   2. **Motion Decomposition:**
>     a)	Move to the red apple.
>     b)	Pick up the apple and place it on the blue ellipse.
>     c)	Move to the yellow banana.
>     d)	Pick up the banana and place it on the blue ellipse.
>     e) Move to the can.
>     f) Pick up the can and place it on the red ellipse.
>    3. **Description:**  This task is a variant of the Table Bussing task. The primary objective is to evaluate the language steerability of MoTVLA when provided with a reversed instruction prompt (from garbage  others to others  garbage), while the visual information remains largely similar. As before, the robot is not explicitly informed which objects are garbage and which are not; instead, MoTVLA must infer this information and complete the task autonomously. The task is determined successful when all the objects are placed to the correct position within fixed steps and failure otherwise.
>
> The following table presents the performance comparison of MoTVLA against the other three baselines on the aforementioned three tasks. All experiments are conducted under the same training settings and evaluated over 50 unseen seeds, with object positions and noise distributions of diffusion policy randomly initialized.
>
> | Model | Tool Pull & Place | Table Bussing| Table Bussing Reverse |
> |:------------|:-----------:|:------------:|:--------:|
> | DP | 0.50 | 0.10 | 0.60 |
> | $\pi$0 | 0.08| 0.54 | 0.50 |
> | $\pi$0.5 KI | 0.10 | 0.26 | 0.70 |
> | MoTVLA | **0.72** | **1.0** | **0.98** |
>
> It is evident from this table that as tasks require deeper internal reasoning, the performance margin of MoTVLA over other baselines becomes larger. For example, in the Table Bussing task, MoTVLA achieves a 46% higher success rate than the second-best baseline, $\pi$0, and outperforms the SOTA $\pi$0.5 KI by up to 28% on the Table Bussing (Reverse) task, confirming MoTVLA’s enhanced capability in long-horizon task performance. The corresponding demonstration video can be found in:  [Click Me! Tool Pull & Place](https://drive.google.com/file/d/1guNmHbDjpkuCnn4HJo2THY1soK46G9TN/view); &emsp; [Click Me! Table Bussing](https://drive.google.com/file/d/17vXHEUtTpo3aoK9P_k3rJ6upkHbRtvrM/view);  &emsp; [Click Me! Table Bussing Reverse](https://drive.google.com/file/d/1W0OVgwimJaNy6red0OEuypsHwCB8fJ2P/view)
>
> We will include these additional quantitative results in the revised manuscript and hope above discussion can well address reviewer's concern.

---

> ### Author Response · Authors · 2025-11-22
> **Response to Reviewer NduJ (IV)**
>
> ### **A2 (Continued): The paper lacks both motivation and ablation studies. It does not verify why and how general and robotic-domain reasoning improve manipulation performance. Which aspect of manipulation is enhanced (e.g., fine-grained control, long-horizon planning, or generalization)? How can the authors justify that preserving science or other general reasoning abilities helps manipulation? This contradicts common understanding in robotics, where retaining such reasoning only increases parameter redundancy and inference latency for real-time control systems. Finally, during policy inference, the DiT head relies solely on fast reasoning features (plus visual observations, etc.) and does not use outputs from slow reasoning as conditions.**
>
> **1. Which aspect of manipulation is enhanced? (Responded in previous two threads.)**
>
> **2. How can the authors justify that preserving science or other general reasoning abilities helps manipulation?**
> This has already been verified in our ablation studies, as shown in **Table 3 Ablation Study of Architecture Design** of our manuscript, where we compare MoTVLA with MoTVLA (scratch), in which the generalist is randomly initialized and therefore lacks general reasoning ability. This experiment investigates the significance of general intelligence.
>
> Nevertheless, we acknowledge that the previous version of the table may not have presented this information clearly, and we have therefore reorganized it below for improved clarity.
>
> | Attribute | Attribute | Bagel w\ DP | MoTVLA (scratch) | MoTVLA (ours) |
> |:------|:--------|:--------:|:--------:|:--------:|
> | Component | Generalist (Slow Reasoning) | Pre-trained | Scratch | Pre-trained |
> |  | Domain Expert (Fast Reasoning) | N/A | Fine-tuned | Fine-tuned |
> |  | Action Expert (Diffusion Policy) | Fine-tuned | Fine-tuned | Fine-tuned |
> | Slow Reasoning| ScienceQA (TA) | 94.12 | 0.0 | 94.12 |
> |  | P&P Motion Decomposition (TA) | 1.68 | 0.0 | 1.68 |
> | Fast Reasoning| ScienceQA (TA) | N/A | 17.00 | **90.99** |
> |  | P&P Motion Decomposition (TA) | N/A | 12.08 | **82.41** |
> | Diffusion Policy | Cube Stacking (SR) | 0.0 | 0.0 | **78.0** |
>
> As shown in this table, MoTVLA (scratch) fails to learn across all tasks under both slow and fast reasoning, confirming that the general intelligence inherited from the pre-trained VLM is essential for effective multimodal reasoning. Consequently, MoTVLA (scratch) achieves zero success on manipulation tasks, as unstable and hallucinated motion decomposition introduces significant variance and disrupts policy learning. On the contrary, our MoTVLA achieves the best performance across all three tasks, confirming the necessity of reserving the general intelligence. This [Plot(Click Me! )](https://drive.google.com/file/d/1RxSStmZWjPvfvWix-ErtNKPGPsokVLKT/view) presents a concrete rollout example with the corresponding motion decomposition. It is evident that the improper and hallucinated reasoning in both baselines causes them to fail to achieve the task goal until the final motion step.
>
> **Note: To be continued in next thread due to characters limitation**

---

> ### Author Response · Authors · 2025-11-22
> **Response to Reviewer NduJ (V)**
>
> ### **A2 (Continued): The paper lacks both motivation and ablation studies. It does not verify why and how general and robotic-domain reasoning improve manipulation performance. Which aspect of manipulation is enhanced (e.g., fine-grained control, long-horizon planning, or generalization)? How can the authors justify that preserving science or other general reasoning abilities helps manipulation? This contradicts common understanding in robotics, where retaining such reasoning only increases parameter redundancy and inference latency for real-time control systems. Finally, during policy inference, the DiT head relies solely on fast reasoning features (plus visual observations, etc.) and does not use outputs from slow reasoning as conditions.**
>
> **1. Which aspect of manipulation is enhanced? (Responded in previous threads.)**
>
> **2(Continued). How can the authors justify that preserving science or other general reasoning abilities helps manipulation?**
>
> While it is possible to fine-tune the slow reasoning of Bagel on our motion decomposition dataset to improve accuracy, this approach would inevitably lead to catastrophic forgetting of general intelligence. To confirm this, we also conducted another ablation study that we fine-tuned the generalist over our robotic datasets and we benchmark both original and fine-tuned generalists (generalist*) on the Massive Multi-discipline Multimodal Understanding and Reasoning (MMMU) dataset [1], following the reasoning benchmark employed in the original Bagel paper. The quantitative result is shown in this [bar plot (Click Me!)](https://drive.google.com/file/d/13F_mSaDbFc2GtRzM-iRhKI7wkDHxinQs/view).
>
> As shown in this figure, after fine-tuning, the performance of slow reasoning degrades across several subjects. For instance, while the tuned generalist maintains its performance in domains such as art, accounting, finance, and other science-related subjects, it exhibits knowledge forgetting in music, marketing, energy and power, and mechanical engineering. This degradation is even more pronounced in public health, where accuracy drops by 6.6% after fine-tuning. We also observe similar catastrophic forgetting in robotic VQA when providing a meta-task description followed by a general question, for example: “Your meta task is: Put the cucumber into the box. Before executing the task, tell me what is the color of the cucumber.” The fine-tuned generalist fails to respond to the question about color and instead outputs the decomposed motion. These results highlight the superiority of MoTVLA, which preserves slow reasoning capability even after robotic VQA learning, confirming the necessity of the MoT architecture for maintaining fast–slow reasoning performance.
>
> We will include the above discussion in the revised manuscript and hope it adequately addresses the reviewer’s concern regarding the significance of preserving general intelligence.
>
> **3. Finally, during policy inference, the DiT head relies solely on fast reasoning features (plus visual observations, etc.) and does not use outputs from slow reasoning as conditions.**
>
> We would like to emphasize that the action expert is intentionally conditioned only on the fast reasoning features, **as these features already attend to those of slow reasoning at each layer through shared global attention mechanisms during the forward process.** The rationale behind this design is that, instead of constructing a hierarchical architecture to separate fast and slow reasoning, we adopt the MoT architecture to connect each attention layer, allowing the fast reasoning to extract both high- and low-level representations from the generalist in a unified manner. The empirical results further confirm the feasibility and effectiveness of this design across both short- and long-horizon tasks. Additionally, incorporating the slow reasoning features as a condition for the action expert would further slow down the inference speed due to the next-token prediction, which contradicts our primary motivation of maintaining general intelligence while performing fast reasoning for motion decomposition to ensure efficient subsequent task execution.
>
> We hope all above discussions could well address the reviewer's concerns.
>
> #### **Reference**
> 1. Yue, Xiang, et al. "Mmmu: A massive multi-discipline multimodal understanding and reasoning benchmark for expert agi." Proceedings of the IEEE/CVF Conference on Computer Vision and Pattern Recognition. 2024.

---

> ### Author Response · Authors · 2025-11-22
> **Response to Reviewer NduJ (VI)**
>
> ### **A3: What is the real innovation of the MoT design compared with Bagel (Deng et al., 2025a)? Is it merely distinguishing general-domain and robotic-domain datasets based on Bagel’s MoT structure and then adding a DiT policy head? Furthermore, the use of bidirectional token-wise reasoning is not motivated by characteristics of robotic tasks.**
>
> Thank you for the reviewer’s questions.We would like to emphasize once again that, as clearly stated in the Introduction section, the core contribution of this work lies in achieving unified fast–slow reasoning based on the MoT architecture, which substantially improves the efficiency of robotic tasks while providing intermittent interpretability within the language context. Although we modified parts of the MoT architecture for our reasoning tasks, for example, by adapting the input modality, adjusting the output decoder of the second transformer, introducing the additional action expert, and redesigning the overall training and inference pipeline to realize slow–fast reasoning with action diffusion, we did not list these as primary contributions, as the basic code framework is derived from Bagel.
>
> Therefore, we do not consider it critical to distinguish architectural innovations relative to Bagel, since MoTVLA is specifically designed for robotic applications, particularly as a VLA model, whereas Bagel is designed for general reasoning and generation tasks. It is evident that the two models belong to different domains, despite both employing the same MoT concept. Furthermore, we would like to clarify that the MoT concept itself predates Bagel, having been originally proposed by Meta [1] to handle multi-modal inputs and outputs.
>
> Regarding the reviewer’s comment on bidirectional token-wise reasoning, we kindly ask for further clarification or additional details so that we can better understand and address the concern. We would be very happy to provide a more detailed explanation or supplementary discussion once the reviewer’s specific points are clarified.
>
> #### **Reference**
> 1.	Liang, Weixin, et al. "Mixture-of-transformers: A sparse and scalable architecture for multi-modal foundation models." arXiv preprint arXiv:2411.04996 (2024).
>
> ---
>
> ### **A4: As a robotic VLA paper, the model’s inference speed should be compared with more VLA baselines such as Fast [1], Pi_0, and CogACT [2]. Although these models do not explicitly generate language, the latent features provided by their LLMs also serve as high-level conditions for action generation. [1] Fast: Efficient Action Tokenization for Vision-Language-Action Models [2] CogACT: A Foundational Vision-Language-Action Model for Synergizing Cognition and Action in Robotic Manipulation.**
>
> Thank you for raising this suggestion. Although MoTVLA is also a VLA model, there exists a critical difference between our approach and the VLAs mentioned by the reviewer. Specifically, both $\pi$0 (and FAST) and CogACT treat the latent features provided by LLMs through one time forward process as conditions for their action generation. However, these features contain only the encoded representations of visual and textual inputs, rather than reasoning-related features, as these models do not generate language through next-token prediction (only the features produced via next-token prediction contain explicit reasoning information).
>
> We would like to point out that this design leads to a major limitation: such VLAs struggle to handle ambiguous instruction prompts. For example, consider a scenario where a tool and a cube are placed on a table, with the cube located beyond the reach of the manipulator. If the user issues the prompt “Place the cube into the box,” the latent features produced by an LLM in a single forward pass would only encode the input information. Since the manipulator cannot reach the cube directly, it must first pick up the tool and use it to pull the cube closer. Clearly, $\pi$0, FAST, and CogACT lack this kind of intermediate reasoning, as they do not perform multi-step reasoning through multiple forward passes (i.e., next-token prediction). Consequently, their action generation is not conditioned on decomposed intermediate motions, making it difficult for them to handle ambiguous instruction prompts.
>
> In contrast, the latent features generated by MoTVLA within a single forward pass already contain all the necessary information about the decomposed motions. This is achieved through the token-wise prediction paradigm, where the domain expert attends to the generalist via global attention mechanisms, enabling the generated features to be directly decoded into a complete motion sequence through the vocabulary dictionary. As a result, MoTVLA substantially outperforms other VLAs in both qualitative results (Figure 4 in our manuscript) and quantitative results (discussed in our response to A2) for long-horizon tasks with ambiguous prompts.
>
> **Note: To be continued in the next thread due to characters limit.**

---

> ### Author Response · Authors · 2025-11-22
> **Response to Reviewer NduJ (VII)**
>
> ### **A4 (Continued): As a robotic VLA paper, the model’s inference speed should be compared with more VLA baselines such as Fast [1], Pi_0, and CogACT [2]. Although these models do not explicitly generate language, the latent features provided by their LLMs also serve as high-level conditions for action generation. [1] Fast: Efficient Action Tokenization for Vision-Language-Action Models [2] CogACT: A Foundational Vision-Language-Action Model for Synergizing Cognition and Action in Robotic Manipulation.**
>
> While we acknowledge that comparing the overall VLA inference speed with other baselines would be beneficial, we found it extremely difficult to make a fair comparison, as different models employ different types of action decoders. For example, Pi0 adopts the flow-matching paradigm for its action expert, whereas Fast tokenizes continuous action signals into discrete symbols. Moreover, Pi0 is implemented using JAX, which is inherently faster at the programming language level, even faster than Diffusion Policy [1], which is implemented in PyTorch.
>
> It is also worth noting that the core idea of MoTVLA is not constrained by the choice of action expert. The current diffusion policy can be flexibly replaced with either a flow-matching or discrete action-tokenization approach. Naturally, the inference would be faster if MoTVLA were implemented in JAX; however, such optimization lies outside the current scope of this work.
>
> Returning to the topic of inference latency, we provide an inference latency comparison against CogACT, which employs the same type of action encoder as MoTVLA, and OpenVLA, an early-stage well-known closed-loop VLA.
>
> | GPU A6000 | MoTVLA | CogACT [2] | OpenVLA [3] |
> |:------------|:-----------:|:------------:|:--------:|
> | Inference Time (ms) | 252 | 181 |307 |
> |Denoising Steps | 50 | Unkown | N/A |
> | Number of generated actions | 16 | 16 | 1 |
>
> Although the comparison is not entirely fair due to differences in parameter sizes and denoising steps, we aim to provide a relevant intuition regarding the inference latency among VLAs.
>
> Thank you once again for raising this point and we hope this explanation clarifies your concerns.
>
> #### **Reference**
> 1. Chi, Cheng, et al. "Diffusion policy: Visuomotor policy learning via action diffusion." The International Journal of Robotics Research 44.10-11 (2025): 1684-1704.
> 2. Li, Qixiu, et al. "Cogact: A foundational vision-language-action model for synergizing cognition and action in robotic manipulation." arXiv preprint arXiv:2411.19650 (2024).
> 3. 2. Kim, Moo Jin, et al. "OpenVLA: An Open-Source Vision-Language-Action Model." Conference on Robot Learning. PMLR, 2025.

---

> ### Author Response · Authors · 2025-11-25
> **Update to Reviewer NduJ**
>
> We have uploaded the revised version of the PDF file, and we would be more than happy to hear any further feedback or engage in discussion with the reviewer to further enhance the quality of our work!

---

### Official Review · Reviewer_NWp5 · 2025-11-01

**Soundness:** 2
**Presentation:** 3
**Contribution:** 3
**Rating:** 6
**Confidence:** 3

**Summary:**

In this paper, the authors address two key challenges in existing Vision-Language-Action models for robot learning: limited language steerability without reasoning and high inference latency with reasoning. They propose MoTVLA, a MoT-based VLA model integrating unified fast-slow reasoning with behavior policy learning. MoTVLA includes a generalist, a domain expert, and an action expert. Its effectiveness is validated via NLP benchmarks, robotic simulations, and real-world tasks, outperforming SOTA baselines.

**Strengths:**

1. This paper proposes a novel MoT-based unified fast-slow reasoning framework, addressing the critical trade-off between language steerability and inference latency in existing VLA models. The "generalist-domain expert" design with knowledge sharing is conceptually sound, filling a gap in current VLA research.

2. To ensure both general intelligence retention and policy efficiency, the authors adopt a "decomposition-composition-decomposition" p
ipeline for reasoning and a two-stage training recipe. This methodological rigor lays a solid foundation for MoTVLA’s performance.

3. Extensive experiments across simulation, real-world, and reasoning tasks corroborate MoTVLA’s superiority in reasoning accuracy, manipulation success rate, and inference latency, providing strong empirical support.

**Weaknesses:**

1. The real-world dataset used for training the action expert is excessively small and relies heavily on manual annotation, which poses significant challenges for data scaling. This not only increases the risk of overfitting but also restricts the model to a narrow range of action skills (e.g., simple pick-and-place, stacking, and pulling tasks). Consequently, the model’s reliability and scalability in diverse real-world scenarios are compromised.

2. The results of reasoning tasks (Section 4.2) lack comparative analyses with SOTA VLMs, making it difficult to accurately assess the competitive advantage of MoTVLA in terms of reasoning capabilities.

3. The paper focuses exclusively on simple short-horizon tasks (e.g., cube stacking) and lacks evaluations on complex long-horizon tasks (with only one qualitative result provided) or tasks involving multi-arm systems or cross-embodiment scenarios. This limitation undermines the generalizability of the MoTVLA framework.

**Questions:**

1. Table 4 contains a mislabeling issue: its title claims to present "dataset composition," while the actual content displays "reasoning latency" data, which may cause confusion for readers. Will the authors correct the label of Table 4 to align with its content?

2. Could the authors add comparative analyses of slow reasoning performance between MoTVLA and SOTA VLMs (e.g., Qwen2.5-VL)? Additionally, would it be possible to supplement quantitative results comparing MoTVLA with baselines on multi-stage (semi-long-horizon) tasks?

3. How does MoTVLA handle unseen skills or complex multi-step instructions in open-world scenarios? Could the authors provide quantitative or qualitative results to demonstrate the model’s performance in such cases?

4. There is an inconsistency in the description of motion merging: the appendix mentions merging the "picking up" and "moving toward the destination" motions, while Figure 4 illustrates the merging of "picking up" and "placing into" motions. Could the authors explain the reason for this discrepancy, clarify the criteria used to determine which motions to merge, and elaborate on how consistency is maintained throughout the motion decomposition process?

5. The paper lacks detailed descriptions of the random states and seeds employed in the experiments. Could the authors supplement this information, including the specific values of the random seeds used and the methods by which initial object poses and robot initial states were randomized?

6. To fully validate the language steerability of MoTVLA, could the authors add verification experiments where the model is instructed to grasp different objects via varying language prompts in the same scene? For example, in a scene containing eggplants, carrots, and corn, the model should perform pick-and-place operations on the target object specified by each distinct instruction.

7. Regarding the identified limitation that MoTVLA exhibits "strong reasoning ability but insufficient execution capability," have the authors attempted to transfer the decomposed sub-instructions (generated in the second stage) to other VLA policies (e.g., π0.5 KI) for execution? If such attempts have been made, what results were obtained? If not, what are the reasons for not pursuing this approach?

---

> ### Author Response · Authors · 2025-11-22
> **Response to Reviewer NWp5 (I)**
>
> We thank the Reviewer NWp5 for appreciating the novelty and experimental solidness of our work. The reviewer raises concerns about the overfitting, reasoning comparisons, and long-horizon tasks, and we have carefully gone through all the comments and responded point by point with additional experiments for necessary support.
>
> ---
>
> ### **A1: Table 4 contains a mislabeling issue: its title claims to present "dataset composition," while the actual content displays "reasoning latency" data, which may cause confusion for readers. Will the authors correct the label of Table 4 to align with its content?**
>
>  Thanks a lot for the suggestion! We corrected the caption to “Comparison of reasoning latency” and will correct it in the revised manuscript.
>
> ---
>
> ### **A2: Could the authors add comparative analyses of slow reasoning performance between MoTVLA and SOTA VLMs (e.g., Qwen2.5-VL)? Additionally, would it be possible to supplement quantitative results comparing MoTVLA with baselines on multi-stage (semi-long-horizon) tasks?**
>
>  Thank you for the valuable suggestions. To thoroughly address the aforementioned concerns, we have conducted additional experiments to further demonstrate the superiority of the proposed MoTVLA in both slow reasoning and multi-stage (long-horizon) tasks.
>
> #### **+ Slow Reasoning Comparison**
>
> The following table presents a performance comparison of slow reasoning across multiple reasoning tasks. Specifically, the metrics for MME-S, MMBench, MMMU, MM-Vet, and MathVista are referenced from Bagel[1], while we additionally evaluate the models on CODA-LM [2] to provide further validation.
>
> | Model | MMBench | MMMU | MM-Vet | MathVista |CODA-LM|
> |:------------|:-----------:|:------------:|:-------:|:--------:|:--------:|
> | LLaVA-OV 7B | 80.8 | 48.8 | 57.5 | 63.2 | 1.3 |
> | QwenVL 2.5 7B | 83.5 | **58.6** | 67.1 | 68.2 | 5.1 |
> | MoTVLA VLM 7B | **85.0** | 55.3 | **67.2** | **73.1** | **6.4** |
>
> It is clear that the MoTVLA outperforms the other two well-known VLMs (LLaVA-OV 7B and QwenVL VLM 7B) on four tasks among five, confirming its SOTA performance of MoTVLA on slow reasoning.
> #### **Reference**
> [1] Deng, Chaorui, et al. "Emerging properties in unified multimodal pretraining." arXiv preprint arXiv:2505.14683 (2025).
> [2] Li, Yanze, et al. "Automated evaluation of large vision-language models on self-driving corner cases." CoRR (2024).
>
> ---
>
> **Note: The response is continued in next thread due to characters limitation**

---

> ### Author Response · Authors · 2025-11-22
> **Response to Reviewer NWp5 (II)**
>
> ### **A2 (continued): Could the authors add comparative analyses of slow reasoning performance between MoTVLA and SOTA VLMs (e.g., Qwen2.5-VL)? Additionally, would it be possible to supplement quantitative results comparing MoTVLA with baselines on multi-stage (semi-long-horizon) tasks?**
>
> #### **+ Quantitative results on multi-stage tasks**
> We appreciate the reviewer’s insightful suggestions regarding the long-horizon tasks, as such experiments further emphasize the significance of motion decomposition and demonstrate the language steerability of MoTVLA. To address this, we have conducted three long-horizon experiments as follows:
>
>  **Tool Pull and Place:**
>
>   1. **Prompts**: Your meta task is: Put the cube into the bin.
>   2. **Motion Decomposition:**
>      a) Move to the position of the L shaped tool on the table.
>      b) Pick up the L shaped tool from the table.
>      c) Use the L shaped tool to pull the cube.
>      d) Move to the cube and pick it up.
>      e) Place the cube into the bin.
>    3. **Description:**  In this task, the primary objective of MoTVLA is to use the tool on the table to pull a cube that is initially out of the reach of the manipulation arm and place it into the designated destination. The key challenge lies in the fact that, instead of providing a detailed instruction, only a simple prompt is given, i.e., “Put the cube into the bin,” which contains no information about the use of the tool. Consequently, the manipulation arm must infer and learn how to utilize the tool by following the motion decompositions. The task is considered successful if the cube is placed in the correct position within a fixed number of steps; otherwise, it is regarded as a failure.
>
>  **Table Bussing:**
>
>   1. **Prompts**: Your meta task is: Place the garbage on the red ellipse and put all the other objects on the blue ellipse. (Note: red/blue ellipse are the markers)
>   2. **Motion Decomposition:**
>     a)	Move to the can.
>     b)	Pick up the can and place it on the red ellipse.
>     c)	Move to the yellow banana.
>     d)	Pick up the banana and place it on the blue ellipse.
>     e)	Move to the red apple.
>      f) Pick up the apple and place it on the blue ellipse.
>    3. **Description:**  In this task, the primary objective of MoTVLA is to clear the objects on the table and classify them into their corresponding destinations. The key challenge lies in the fact that the user does not specify which objects are garbage and which are not. The robot must therefore infer this information autonomously and complete the task by following the decomposed motions rapidly inferred by MoTVLA. The task is considered successful if all objects are placed in their correct positions within a fixed number of steps and considered as a failure otherwise.
>
>  **Table Bussing Reverse:**
>
>   1. **Prompts**: Your meta task is: Place all the fruits on the blue ellipse and put the garbage on the red ellipse.
>   2. **Motion Decomposition:**
>     a)	Move to the red apple.
>     b)	Pick up the apple and place it on the blue ellipse.
>     c)	Move to the yellow banana.
>     d)	Pick up the banana and place it on the blue ellipse.
>     e) Move to the can.
>     f) Pick up the can and place it on the red ellipse.
>    3. **Description:**  This task is a variant of the Table Bussing task. The primary objective is to evaluate the language steerability of MoTVLA when provided with a reversed instruction prompt (from garbage  others to others  garbage), while the visual information remains largely similar. As before, the robot is not explicitly informed which objects are garbage and which are not; instead, MoTVLA must infer this information and complete the task autonomously. The task is determined successful when all the objects are placed to the correct position within fixed steps and failure otherwise.
>
> The following table presents the performance comparison of MoTVLA against the other three baselines on the aforementioned three tasks. All experiments are conducted under the same training settings and evaluated over 50 unseen seeds, with object positions and noise distributions of diffusion policy randomly initialized.
>
> | Model | Tool Pull & Place | Table Bussing| Table Bussing Reverse |
> |:------------|:-----------:|:------------:|:--------:|
> | DP | 0.50 | 0.10 | 0.60 |
> | $\pi$0 | 0.08| 0.54 | 0.50 |
> | $\pi$0.5 KI | 0.10 | 0.26 | 0.70 |
> | MoTVLA | **0.72** | **1.0** | **0.98** |
>
> It is evident that MoTVLA significantly outperforms the other baselines by a large margin across all three tasks. These results emphasize the importance of motion decomposition (fast reasoning) for language-steerable behaviors, particularly in long-horizon tasks with ambiguous instructions that require strong reasoning capabilities.
>
> We sincerely thank the reviewer once again for the valuable comments. We will include these new quantitative results in the revised manuscript.

---

> ### Author Response · Authors · 2025-11-22
> **Response to Reviewer NWp5 (III)**
>
> ### **A3: How does MoTVLA handle unseen skills or complex multi-step instructions in open-world scenarios? Could the authors provide quantitative or qualitative results to demonstrate the model’s performance in such cases?**
>
> Thanks for the valuable comments. We agree that richer motion skills are required to handle open-world scenarios. Due to high cost of motion decomposition annotations and curation, the current MoTVLA primarily focuses on tabletop tasks. However, we would like to emphasize that this is not an inherent limitation of MoTVLA but rather a constraint related to data scaling and hardware conditions. As a future research direction, we plan to further generalize MoTVLA to a broader range of open-world tasks by exploring motion and skill decompositions as well as their compositions, to enable the emergence of new behaviors in unseen scenarios.
>
> Regarding the multi-step instructions, we have conducted three additional long-horizon tasks, which are discussed in our response to Q2. We refer the quantitative results provided in our response to A2 and attach the corresponding qualitative results in this thread.
> **Demonstration Video:** [Click Me! Tool Pull & Place](https://drive.google.com/file/d/1guNmHbDjpkuCnn4HJo2THY1soK46G9TN/view); &emsp; [Click Me! Table Bussing](https://drive.google.com/file/d/17vXHEUtTpo3aoK9P_k3rJ6upkHbRtvrM/view);  &emsp; [Click Me! Table Bussing Reverse](https://drive.google.com/file/d/1W0OVgwimJaNy6red0OEuypsHwCB8fJ2P/view)
>
> ---
> ### **A4: There is an inconsistency in the description of motion merging: the appendix mentions merging the "picking up" and "moving toward the destination" motions, while Figure 4 illustrates the merging of "picking up" and "placing into" motions. Could the authors explain the reason for this discrepancy, clarify the criteria used to determine which motions to merge, and elaborate on how consistency is maintained throughout the motion decomposition process?**
>
> This is a great question!
> The motion merging strategy primarily depends on the type of task. For pick-and-place tasks, we omit the “move toward” motion and merge “picking up” and “placing into,” since “placing into” inherently contains the semantic meaning of “move toward.” Otherwise, we found “placing into” would represent only a very short motion segment that contains extremely small number of data samples. In contrast, for tasks involving more distinct terminating motions, such as “inserting the peg into a hole” or “pulling a cube with a tool”, we keep “move toward” motion and merge it with “picking up”, while reserving the final motion independent.
>
> The rationale behind this design is to ensure accurate and semantically consistent annotations while maintaining a relatively balanced number of samples for each motion type. This helps mitigate potential overfitting to any dominated single motion pattern.
> We hope this explanation clarifies the reviewer’s questions, and we will include this discussion in the appendix of the revised manuscript.
>
> ---
> ### **A5: The paper lacks detailed descriptions of the random states and seeds employed in the experiments. Could the authors supplement this information, including the specific values of the random seeds used and the methods by which initial object poses and robot initial states were randomized?**
>
> Thanks to the reviewer for this valuable suggestion. All training samples were collected using seeds ranging from 0 to the total number of collected trajectories (e.g., seeds 0~300 for the Peg-in-Hole task), while all inference seeds ranged from 1000 to 1049. Each seed triggers randomization of the object positions on the table as well as the initialization of Gaussian noise for the diffusion policy through the interface functions in our codebase. We will include this clarification in the appendix of the revised manuscript and hope it adequately addresses the reviewer’s question.
>
> ---
> ### **A6: To fully validate the language steerability of MoTVLA, could the authors add verification experiments where the model is instructed to grasp different objects via varying language prompts in the same scene? For example, in a scene containing eggplants, carrots, and corn, the model should perform pick-and-place operations on the target object specified by each distinct instruction.**
>
> We thank for this suggestion and are conducting verification experiments on this question. We would post reply very soon for this in an individual thread.

---

> ### Author Response · Authors · 2025-11-22
> **Response to Reviewer NWp5 (IV)**
>
> ### **A7: Regarding the identified limitation that MoTVLA exhibits "strong reasoning ability but insufficient execution capability," have the authors attempted to transfer the decomposed sub-instructions (generated in the second stage) to other VLA policies (e.g., π0.5 KI) for execution? If such attempts have been made, what results were obtained? If not, what are the reasons for not pursuing this approach?**
>
> Thank you for this valuable comment.
> In fact, we have attempted to integrate a well-trained action expert, such as the one from $\pi$0.5 KI, with our reasoning backbone. However, we found that this integration cannot be achieved in a zero-shot manner, as the interface between the domain expert and the action expert in MoTVLA differs significantly from that in Pi0.5 KI. According to the official GitHub repository of Pi0.5 KI, the currently released code only supports the action head. As a result, the sub-task reasoning is deeply intertwined with their action expert, making it difficult to decouple the action head and reassemble it within MoTVLA.
>
> As a future research direction, we plan to train MoTVLA as a fully end-to-end VLA model without any intermediate reasoning tasks in Stage 1, which would allow us to efficiently leverage more action-related open-source datasets. We will then fine-tune the action expert together with the domain expert on our annotated reasoning and action tasks, enhancing the language steerability along with the open-world task performance.

---

> ### Author Response · Authors · 2025-11-25
> **Update to Reviewer NWp5**
>
> We have uploaded the revised version of the PDF file, and we would be more than happy to hear any further feedback or engage in discussion with the reviewer to further enhance the quality of our work!

---

> ### Author Response · Authors · 2025-11-25
> **Update regarding A6**
>
> ### **A6: To fully validate the language steerability of MoTVLA, could the authors add verification experiments where the model is instructed to grasp different objects via varying language prompts in the same scene? For example, in a scene containing eggplants, carrots, and corn, the model should perform pick-and-place operations on the target object specified by each distinct instruction.**
>
> To evaluate the language steerability of MoTVLA, we tested a checkpoint trained on scenes containing only a single object (either an apple or a banana). During inference, we placed both objects together and provided object-specific prompts to assess whether the model could correctly follow the instructions. For each test prompt, we conducted 20 trials using unseen seeds. The results are shown in the following table:
>
> | Metric| Prompt: Put the red apple on the white ellipse. | Prompt: Put the yellow banana on the white ellipse.|
> |:------------|:-----------:|:--------:|
> | Success Rate| 0.90 | 0.95 |
>
> Although the visual zero-shot setup prevents MoTVLA from achieving perfect performance, the relatively high success rates demonstrate its strong language steerability, successfully picking and placing the instructed objects based on the given prompts.
>
> The qualitative results can be found at the following links: [Click Me! Banana](https://drive.google.com/file/d/1-kt53ZN8kCSaLqWC9FjYbKAreL-k0IL1/view?usp=drive_link); [Click Me! Apple](https://drive.google.com/file/d/1Vr6K3XVEskbUMAwAA-vYxgSTFVt7dB1g/view?usp=drive_link)
>
> We have included this additional study in the appendix (A.6 Language Steerability of MoTVLA.) of the revised manuscript, and we hope that this response could well address reviewer’s concern.

---

### Author Response · Authors · 2025-12-02
**Overall Rebuttal Summary**

**Dear Reviewers, Area Chairs, and Program Chairs,**

While we appreciate the reviewers’ time and efforts, we were concerned that the discussion phase concluded before the usual timeline and that we did not receive any follow-up responses despite having submitted our rebuttal and clarifications early. **We invested substantial effort in addressing all reviewer concerns and suggestions as thoroughly as possible. In particular, we would like to respectfully ask the Area Chair to reconsider the assessment provided by Reviewer NduJ.** The reviewer’s decision to recommend rejection based primarily on concerns about motivation and novelty, without engaging in technical discussion in the initial review or during the rebuttal phase, and without responding to our clarifications, appears inconsistent with the evaluations of the other reviewers, who explicitly acknowledged the contributions and novelty of MoTVLA.

Last but not least, we hope that the clarifications, additional analyses, and newly introduced experiments have adequately addressed the main concerns raised by the Reviewers. We have incorporated all major points discussed during the rebuttal into the revised paper and appendix, with all modifications highlighted in blue for ease of reference. **We respectfully request that the considerations presented during the rebuttal, along with the improvements made in direct response to each comment, be fully taken into account when assessing the merit and contribution of MoTVLA.** We again thank the Reviewers for their time and constructive feedback, and we are grateful to the Area Chairs and Program Chairs for their continued evaluation.

Best regards,

Authors of submission 8534

---

### Author Response · Authors · 2025-12-02
**Overall Rebuttal Summary**

**Dear Reviewers, Area Chairs, and Program Chairs,**

First and foremost, we would like to express our sincere appreciation for your time and thoughtful consideration. We are encouraged by the relatively positive reception of MoTVLA, and we are grateful that the reviews explicitly recognized three core aspects of our work:

**Novelty**

  1. #### "This paper proposes a novel MoT-based unified fast-slow reasoning framework." (**Reviewer NWp5**)
  2. #### "The decomposition–composition–decomposition design and the use of fast motion decomposition to condition diffusion policies is a novel and pragmatic formulation."  (**Reviewer beNN**)

  3. #### "The unified fast–slow reasoning architecture via a Mixture-of-Transformers is a creative synthesis that removes a key limitation in prior VLA systems." (**Reviewer beNN**)

**Methodological rigor and experimental thoroughness:**

  1. #### "This methodological rigor lays a solid foundation for MoTVLA’s performance."  (**Reviwer NWp5**)

  2. #### "Extensive experiments across simulation, real-world, and reasoning tasks providing strong empirical support." (**Reviwer NWp5**)
  3. #### "The paper presents a thorough empirical evaluation across complementary fronts with strong, consistent gains over strong baselines (π0/π0.5 KI, DP, GR-MG) and clear latency advantages." (**Reviewer beNN**)

**Clarity and Significance**

1. #### "The writing is clear, and the paper is exceptionally well organized. The figures are thoughtfully designed, with harmonious colors and a clean, balanced layout." (**Reviewer LEWC**)

2. #### "The model components and inference pipeline are clearly described, supported by pseudo-code, figures, and metric choices that align with the intended behaviors." (**Reviewer beNN**)

**Meanwhile, we have carefully reviewed all comments and provided detailed point-by-point responses addressing both the questions and the identified weaknesses. As the discussion period concluded earlier than anticipated, we summarize below the key points and additional analyses from our rebuttal to facilitate a clear understanding of the updates for the Reviewers and Area Chairs.**

#### **Reviewer NWp5** raised concerns primarily regarding the evaluation of MoTVLA on long-horizon tasks, the need for language-steerability experiments, and comparisons against other state-of-the-art VLMs on reasoning benchmarks. To thoroughly address these points, we conducted extensive new analyses. Specifically, we added quantitative slow-reasoning comparisons between MoTVLA and LLaVA-OV and Qwen2.5-VL across multiple benchmarks—including MMBench, MMMU, MM-Vet, MathVista, and CODA-LM (A2). We also evaluated MoTVLA on three long-horizon manipulation tasks, comparing it with Pi0.5 KI, Pi0, and DP using both quantitative results (A2) and qualitative analyses (A3). In addition, we performed further experiments to validate MoTVLA’s language steerability (A6).

#### **Reviewer beNN** primarily requested ablation studies on long-horizon tasks as well as analyses related to error-recovery behaviors. To comprehensively address these concerns, we conducted an additional ablation study on a representative long-horizon task, Tool Pull & Place, and compared the performance of MoTVLA with MoTVLA trained from scratch, Bagel w/ DP, and Bagel fine-tuned w/ DP. The corresponding qualitative results and detailed analyses are provided in (A2). For the error-recovery behaviors, we included several qualitative video demonstrations along with descriptive explanations in (A3).

#### **Reviewer LEWC** primarily raised questions regarding long-horizon task benchmarks, VQA capability comparisons, and the explanation of inference latency. To appropriately incorporate the reviewer’s suggestions, we benchmarked the performance of MoTVLA on three long-horizon tasks and provided the corresponding analyses (A1). We further evaluated MoTVLA on additional VQA and motion decomposition datasets and reported the quantitative results (A2). In addition, we quantified the inference latency of MoTVLA and compared it with the two VLAs mentioned by the reviewer, CogACT and OpenVLA (A4). It is unfortunate that the discussion with Reviewer LEWC concluded just as it began, but we believe that the extensive additional experiments and corresponding analyses we have provided will allow the reviewer to more fully appreciate the technical contributions of MoTVLA.

#### **Reviewer NduJ** raised several questions concerning long-horizon tasks, the motivation of the framework, and its novelty. We provided detailed responses to Reviewer NduJ, including clarifications of the motivation and key contributions of MoTVLA, as well as additional experiments on long-horizon tasks, ablation studies, and inference latency.

---

### Meta-Review · Area_Chair_GMmb · 2026-01-06

**Summary:**

This paper proposes MoTVLA, a Mixture-of-Transformers architecture designed to unify "slow" high-level reasoning (via a generalist VLM) and "fast" low-level control (via a domain expert). While the "Fast-Slow" unified reasoning concept is novel and theoretically sound, the submission’s primary contention lies in its empirical validation. The initial reviews highlighted a lack of complex, long-horizon tasks. Although the authors added custom long-horizon experiments during the rebuttal, the evaluation suite remains limited to ManiSkill and custom setups, lacking broad, standardized validation.

**Reviewer Concerns:**

Addressed Concerns:

1) Task Complexity: Reviewers (NWp5, LEWC) criticized the initial focus on short-horizon tasks. The authors responded by adding three custom long-horizon tasks (e.g., Tool Pull, Table Bussing), satisfying the specific requests of the reviewers.

2) Baselines & Steerability: The authors successfully added VLM reasoning baselines (Qwen, LLaVA) and demonstrated language steerability/error recovery (responding to beNN and NWp5).

Outstanding Concerns:

1) Lack of Standardized Benchmarks: As noted by the Area Chair, despite the rebuttal improvements, the experimental section remains insufficient for a general VLA paper. The evaluation is missing established, reproducible benchmarks such as LIBERO, SIMPLER, RoboCASA, or RoboTwin. Without these, it is difficult to objectively assess the model's generalization capabilities and "world understanding" against current SOTA beyond specific, potentially narrow custom environments.

2) Motivation: Reviewer NduJ’s fundamental concern regarding the necessity of the complex architecture versus simpler baselines (like Bagel or standard diffusion policies) remains relevant, as the performance gains are not verified on the aforementioned diverse standard suites.

**Reviewer Scores:**

Reviewer NWp5: Likely improved due to the authors' responsiveness on specific requests.

Reviewer beNN: Likely improved due to the inclusion of steerability and error recovery demos.

Reviewer LEWC: May marginally improve due to technical clarifications, but the underlying concern about the "necessity of large models" typically requires stronger benchmarks to fully resolve.

Reviewer NduJ: Unlikely to change, as the core motivation and comparative advantage on standard setups remain unaddressed.

---

### Decision · Program_Chairs · 2026-01-26

Reject